# Monte Carlo Tree Search for Comprehensive Exploration in LLM-Based Automatic Heuristic Design

**Zhi Zheng**[1] **Zhuoliang Xie**[2] **Zhenkun Wang**[2] **Bryan Hooi**[1]

## Abstract

Handcrafting heuristics for solving complex optimization tasks (e.g., route planning and task allocation) is a common practice but requires extensive domain knowledge. Recently, Large Language Model (LLM)-based automatic heuristic design (AHD) methods have shown promise in generating high-quality heuristics without manual interventions. Existing LLM-based AHD methods employ a population to maintain a fixed number of top-performing LLM-generated heuristics and introduce evolutionary computation (EC) to iteratively enhance the population. However, these population-based procedures cannot fully develop the potential of each heuristic and are prone to converge into local optima. To more comprehensively explore the space of heuristics, this paper proposes to use Monte Carlo Tree Search (MCTS) for LLM-based heuristic evolution. The proposed MCTS-AHD method organizes all LLM-generated heuristics in a tree structure and can better develop the potential of temporarily underperforming heuristics. In experiments, MCTS-AHD delivers significantly higher-quality heuristics on various complex tasks. Our code is available[3].

## 1. Introduction

Manually designed heuristics are promising in addressing complex optimization tasks (e.g., combinatorial optimization (CO) problems) (Desale et al., 2015). They are widely used in real-world applications, such as traffic control (He

[1]School of Computing, National University of Singapore, Singapore [2]Guangdong Provincial Key Laboratory of Fully Actuated System Control Theory and Technology, School of Automation and Intelligent Manufacturing, Southern University of Science and Technology, Shenzhen, China. Correspondence to: Bryan Hooi <bhooi@comp.nus.edu.sg>.

*Proceedings of the 42$^{nd}$ International Conference on Machine Learning*, Vancouver, Canada. PMLR 267, 2025. Copyright 2025 by the author(s).

[3]Our code is available at https://github.com/zz1358m/MCTS-AHD-master.

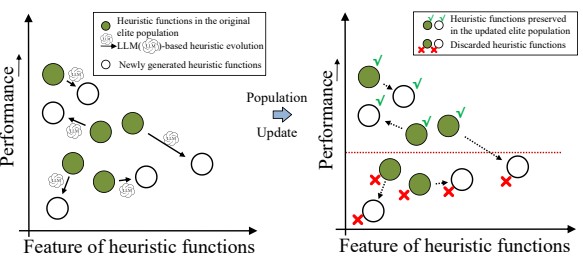

(a) Populations in LLM-based AHD

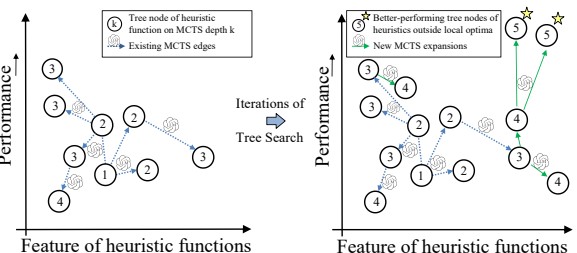

(b) MCTS for higher-quality LLM-based AHD (Ours)

*Figure 1.* The generally adopted population (a) in existing LLM-based AHD methods (Liu et al., 2024b; Ye et al., 2024a) directly discards low-performance heuristics (under the red dashed line in (a)), thus falling into local optima. MCTS provides chances to develop low-performance heuristics, so it can more comprehensively explore the space of heuristic functions with different features.

et al., 2011), job scheduling (Rajendran, 1993), and robotics (Tan et al., 2021). However, manually crafted heuristics often contain intricate workflows and parameter settings, making their design labor-intensive and reliant on task-specific expert knowledge. To achieve easier heuristic design across various tasks, the concept of Automatic Heuristic Design (AHD) (Burke et al., 2013) (also known as Hyper-Heuristics (Ye et al., 2024a)) has attracted extensive attention. It seeks the best-performing heuristic algorithm among valid options. Genetic Programming (GP) (Langdon & Poli, 2013) is commonly employed for AHD, with GP-based AHD methods introducing a series of mutation operators to gradually refine heuristic algorithms (Duflo et al., 2019). Nevertheless, the effectiveness of GP-based methods still relies on human definitions of permissible operators (Liu et al., 2024b), which poses additional implementation difficulties.

In recent years, large language models (LLMs) have shown exceptional effectiveness in various domains (Hadi et al., 2023; 2024; Naveed et al., 2023). Leveraging LLMs to automatically enhance heuristics, LLM-based AHD methods (Liu et al., 2024c; Yu & Liu, 2024) can design high-quality heuristics for various complex tasks without manual interventions. EoH (Liu et al., 2023a; 2024b) and Funsearch (Romera-Paredes et al., 2024) innovatively apply LLMs to AHD by designing a population-based evolutionary computation (EC) procedure. For a given task, several established general frameworks may exist for implementing heuristics, e.g., greedy solving frameworks or frameworks with search-based ideas for CO problems. LLM-based AHD methods focus on designing key heuristic functions within one of these predefined general frameworks, rather than developing heuristics from scratch. These methods maintain a population of outstanding heuristic functions based on their performances on an evaluation dataset and iteratively prompt LLMs to generate new heuristic functions taking the existing ones as starting points. Building on this population-based EC framework, studies also introduce effective components. EoH develops several prompt strategies that guide LLMs in generating effective heuristics. ReEvo (Ye et al., 2024a) incorporates the reflection mechanism (Shinn et al., 2024) to enhance the reasoning of LLMs among heuristic function samples. HSEvo (Dat et al., 2024) presents diversity metrics and harmony search (Shi et al., 2013) to increase the diversity of the population without compromising effectiveness.

These population-based methods eliminate inferior heuristic functions in order to focus more on top-performance ones. However, lower-performance heuristic functions still have the potential to be outstanding after steps of LLM-based refinement. Thus, as shown in Figure 1, the population structure may guide the evolution of heuristics getting stuck in suboptimal local optima. Funsearch and HSEvo employ multiple-population structures (Cantú-Paz et al., 1998) and diversity metrics, respectively, to improve the diversity of populations. Populations with these components can reduce the probability of premature convergence but still fail to explore the complex space of heuristics.

To address these drawbacks, this paper proposes MCTS-AHD, the first tree search method for LLM-based AHD, which employs a tree structure over all the heuristic functions generated so far and uses the Monte Carlo Tree Search (MCTS) algorithm with progressive widening technique (Coulom, 2007) to guide the evolution of further heuristics. Its advantages are: **1)** Instead of directly discarding inferior heuristic functions, MCTS-AHD enables steps of LLM-based evolution on temporarily underperforming heuristic functions while keeping focus on better-performing ones, achieving a more comprehensive exploration of the heuristic space. **2)** The tree structure in MCTS can benefit the LLM-based heuristic refinement, where MCTS-AHD presents a novel tree-special prompt strategy to inspire LLMs with the organized function samples in the MCTS tree paths. Moreover, as components, MCTS-AHD proposes an exploration-decay technique, which linearly decreases the rate at which we perform branching in MCTS as the algorithm progresses. We also present a novel thought-alignment procedure to generate precise linguistic descriptions of functions. In experiments, we implement MCTS-AHD to design heuristics with several general frameworks for a wide range of NP-hard (Ausiello et al., 2012) CO problems and a Bayesian Optimization (BO)-related optimization task. MCTS-AHD achieves significantly higher-quality heuristics than hand-crafted heuristics and existing LLM-based AHD methods.

## 2. Preliminary

### 2.1. Definition: AHD & LLM-based AHD

**AHD:** For a given task $P$ (e.g., a CO problem), AHD methods search for the best-performing heuristic $h^*$ within a heuristic space $H$ as follows:

$$h^* = \arg\max_{h \in H} g(h). \tag{1}$$

The heuristic space $H$ contains all feasible heuristics for tasks $P$. Given a task $P$, *heuristic* $h \in H$ is an algorithm mapping the set of task inputs (also called instances) $I_P$ into the corresponding set of solutions $S_P$, i.e., $h : I_P \to S_P$. For example, heuristics for an NP-hard CO task, Traveling Salesman Problem (TSP), map city coordinates (TSP instances) to travel tours (solutions). The function $g(\cdot)$ is a performance measure function for heuristics, $g : H \to \mathbb{R}$. For a CO task $P$ minimizing an objective function $f : S_P \to \mathbb{R}$, AHD methods estimate the performance function $g$ by evaluating the heuristic $h$ on a task-specific dataset $D$. Formally, we enumerate instances $\boldsymbol{ins} \in D \subseteq I_P$, obtain their solutions $h(\boldsymbol{ins})$ by $h$, and calculate the expectation of the objective function values for the solutions as follows:

$$g(h) = \mathbb{E}_{\boldsymbol{ins} \in D}[-f(h(\boldsymbol{ins}))], \tag{2}$$

The heuristics in $H$ belong to a wide range of general frameworks, and the optimal heuristic could be implemented with multiple frameworks. To focus more on critical heuristic designs within each framework, AHD methods typically predefine a general framework and design only a **key heuristic function** with specified inputs and outputs within this framework. For example, heuristics for TSP within a step-by-step construction framework sequentially build TSP tours one city after another. AHD methods within this framework for TSP will design a key heuristic function to select the next city based on the partial tour of previously visited cities. For simplicity, we still denote the function to be designed as $h$.

**LLM-based AHD:** LLM-based AHD introduces LLMs into the search process for the optimal heuristic function

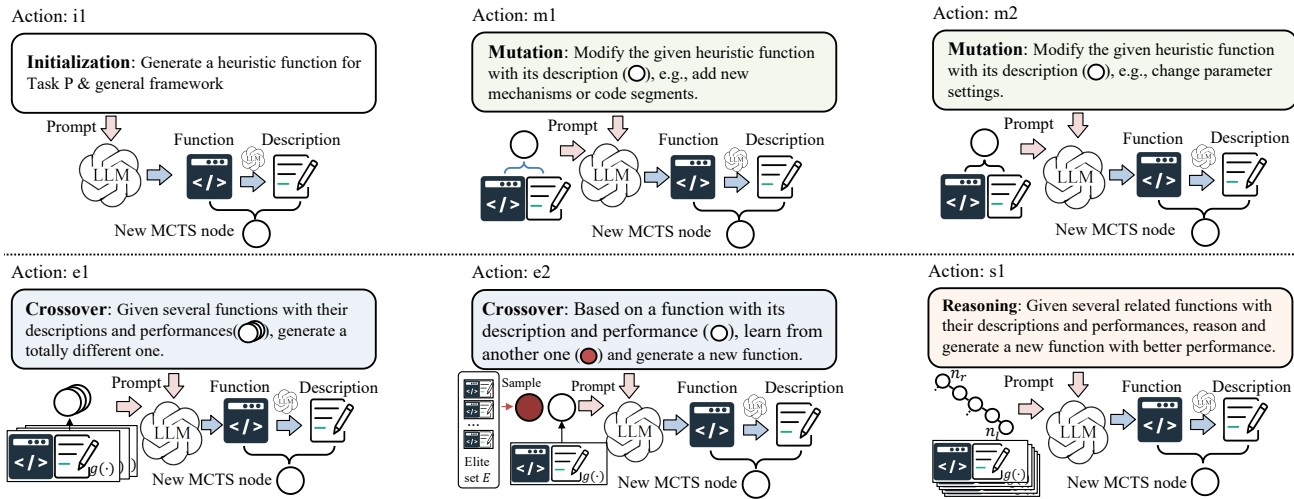

*Figure 2.* LLM-based actions in MCTS-AHD for heuristic evolution. Actions include initializing a new heuristic (i1); two mutation actions (m1 and m2) to mutate an existing heuristic function into a new one with diverse mechanism or detail settings; two crossover actions (e1 and e2) to generate a new heuristic from multiple existing ones; and a novel tree-path reasoning action (s1) to get a better heuristic function from organized function samples on an MCTS tree path from the root node $n_r$ to a leaf node $n_l$.

$h^*$ (within a predefined framework). Existing LLM-based AHD methods (Liu et al., 2024b; Ye et al., 2024a) maintain a population of $M$ heuristic functions $\{h_1, \ldots, h_M\}$ and employ EC to update the population iteratively. Imitating mutation or crossover operators in EC, LLM-based AHD methods prompt LLMs to generate heuristics given existing heuristics from the population. In these methods, a heuristic $h$ will only be retained for the next iteration if its performance $g(h)$ estimated in the evaluation dataset $D$ exceeds the worst-performing heuristic in the original population, i.e., $g(h) > \min_{i \in \{1,\ldots,M\}} g(h_i)$. This property makes it difficult for population-based methods to adopt a worse-before-better strategy for the optimal heuristic $h^*$.

## 2.2. Monte Carlo Tree Search

**MCTS Algorithm.** MCTS (Świechowski et al., 2023) is a decision-making algorithm widely used in games (Silver et al., 2016) and complex decision-making tasks (Fu et al., 2021). Recent studies also verify the power of MCTS in assisting LLMs to conduct multi-hop reasoning (Feng et al., 2023). In the MCTS tree, each node $n_c$ represents a state, and the MCTS will repeatedly select and expand the node with the highest potential judged by a UCT algorithm (Kocsis & Szepesvári, 2006). Each MCTS tree node $n_c$ records a quality value $Q(n_c)$ and a visit count $N(n_c)$ representing the number of times the node has been selected. From an initial root node $n_r$, MCTS gradually builds the MCTS tree to explore the entire state space. Each round of MCTS consists of four stages as follows:

*Selection*: The selection stage identifies the most potential MCTS tree node for subsequent node expansions. From the root node $n_r$, the selection stage iteratively selects the child node with the largest UCT value until it reaches a leaf node. Given a current node $n_c$, the UCT values for its child nodes $c \in \text{Children}(n_c)$ are calculated as follows:

$$\text{UCT}(c) = \left( Q(c) + \lambda \cdot \sqrt{\frac{\ln(N(n_c) + 1)}{N(c)}} \right). \quad (3)$$

*Expansion*: The expansion stage obtains multiple child nodes from the current (leaf) node $n_c$ by sampling several actions from the state of $n_c$ among all possible options.

*Simulation*: The simulation stage evaluates all newly expanded leaf nodes for their quality value $Q(\cdot)$.

*Backpropagation*: The backpropagation process uses the results of simulations to update the values of $Q(\cdot), N(\cdot)$ for nodes on the tree path from $n_c$ to the root $n_r$.

After several iterations of MCTS, the MCTS tree can contain nodes with high-quality states for games or tasks.

**Progressive Widening.** Conventional MCTS is not suitable for tasks with extensive action options or dynamically changing environments (Lee et al., 2020b). Progressive widening (Coulom, 2007) is a technique designed for MCTS to fit such tasks. It gradually adds new child nodes to non-leaf nodes as their visit counts $N(\cdot)$ increases. Formally, for node $n_c$ with child nodes $\text{children}(n_c)$, a new child node will be added every time the following condition is satisfied:

$$\lfloor N(n)^\alpha \rfloor \geq |\text{children}(n_c)|, \quad (4)$$

where $|\text{children}(n_c)|$ is the number of child nodes of $n_c$.

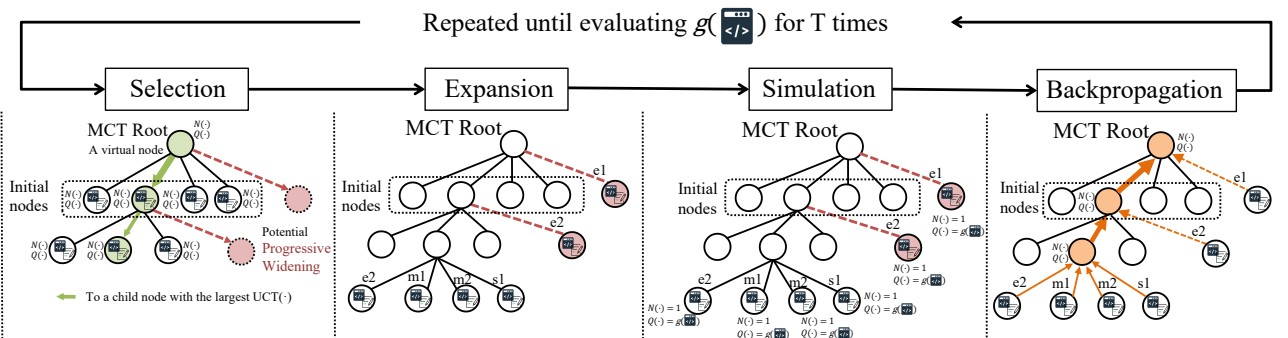

*Figure 3.* The MCTS process in MCTS-AHD contains four stages, i.e., *selection*, *expansion*, *simulation*, and *backpropagation*. MCTS-AHD simulates the quality value of each node as the performance function values of their heuristics and the MCTS will terminate after total $T$ performance evaluations. MCTS-AHD introduces the progressive widening technique to better crossover original heuristic functions with continuously generated new ones. It conducts the crossover actions e1 for the root node and action e2 for other nodes.

## 3. MCTS-AHD

To address the challenges faced by population-based LLM-based AHD methods in overcoming local optima and exploring complex heuristic spaces, this paper proposes a novel method called MCTS-AHD. It preserves all heuristic functions LLMs generated so far in an MCTS tree and employs MCTS with progressive widening for heuristic evolution. The MCTS root node $n_r$ is a virtual node without representing any heuristics, and each of the other nodes represents an executable Python implementation of a heuristic function $h \in H$ along with its linguistic description.

As LLM-based actions for MCTS node expansion, MCTS-AHD prompts LLMs in existing strategies (e.g., mutation, crossover) and a novel tree-path reasoning strategy. Moreover, to encourage the exploration of the heuristic space $H$ in the early stages of MCTS and ensure convergence in the later stages, MCTS-AHD presents an exploration-decay technique that linearly reduces the exploration factor in UCT.

### 3.1. LLM-based actions in MCTS-AHD

To simulate the mutation or crossover operators in EC, LLM-based AHD methods prompt LLMs with several prompt strategies to generate new heuristic functions from existing ones (Liu et al., 2024b). Utilizing the advantage that the MCTS tree can record the relationships of all the generated heuristics, as shown in Figure 2, MCTS-AHD presents a novel MCTS-specific set of actions for LLM-based heuristic evolution, including action i1, e1, e2, m1, m2, and s1. The detailed prompts for these actions are in Appendix E.1. As a commonality, all prompts contain descriptions of the task $P$, the predefined general framework, and the inputs and outputs of the key heuristic function along with their meanings. For actions except i1, prompts also contain implementations and descriptions of existing heuristic functions. For each

action, LLMs are supposed to output the Python code of a heuristic function and its linguistic description.

*Initial Action i1*: As an action for initialization, the action i1 aims to directly generate a heuristic function and its corresponding description from scratch by LLMs.

*Mutation Action m1 & m2*: MCTS-AHD incorporates two mutation actions, m1 & m2, to attempt more detailed designs within the original function workflow. Based on the inputted heuristic function, action m1 prompts LLMs to introduce new mechanisms and formulas, and action m2 prompts LLMs to modify the parameter settings.

*Crossover Action e1*: To explore heuristic functions with new workflows, we employ a crossover action e1 to generate a new heuristic function that diverges from multiple existing ones. These existing heuristic functions are inputted to LLMs with their performances $g(\cdot)$ and descriptions.

*Crossover Action e2*: The crossover action e2 prompts LLMs with a parent heuristic function, a reference heuristic function, and their respective performances. LLMs are supposed to identify the beneficial traits and settings in the reference one and craft an improved heuristic function based on the parent one. The reference is sampled from a dynamically maintained **elite heuristic function set** $E$ consisting of heuristic functions with top 10 performances $g(\cdot)$.

*Tree-path Reasoning Action s1*: The MCTS tree paths from the root $n_r$ to leaf nodes record the evolution history of heuristic functions. So, utilizing this feature, MCTS-AHD presents a novel action s1 to analyze all unique heuristic functions from the leaf node to be developed $n_l$ up to the root $n_r$, identify advantageous designs in these heuristics, and generate an enhanced heuristic function.

**Generating Descriptions: The Thought-Alignment Process**. Linguistic descriptions of heuristic functions can assist the reasoning of LLMs (Liu et al., 2024b), where EoH

prompts LLMs for a description before outputting the function code. However, due to LLM hallucination (Huang et al., 2023), such a procedure could lead to uncorrelated codes and descriptions. For correlated and detailed descriptions, MCTS-AHD proposes a thought-alignment process that summarizes descriptions after the code generation.

Therefore, performing all the MCTS-AHD actions calls LLMs twice. In the first call, we adopt the LLM-generated Python implementation of a heuristic function with action prompts. Then, we conduct a thought-alignment process, prompting LLMs to generate linguistic code descriptions in up to three sentences (refer to Appendix E.2 for prompts). The second LLM call is significantly shorter than the first one, so it will not cause a severe increase in execution time and token cost.

### 3.2. MCTS settings

Figure 3 displays the MCTS process in MCTS-AHD. It first generates $N_I$ initial nodes representing and links them to a virtual root node $n_r$ that does not represent any heuristics. Subsequently, similar to the regular MCTS introduced in Section 2.2, MCTS-AHD repeatedly performs the selection, expansion, simulation, and backpropagation stages as follows until the number of evaluations of heuristics approaches a limit $T$:

*Selection:* In the selection process of MCTS-AHD, MCTS-AHD normalizes the quality value $Q(\cdot)$ to enhance the homogeneity of different tasks in calculating UCT value for child nodes $c \in \text{Children}(n_c)$ of a node $n_c$ as follows:

$$\text{UCT}(c) = \left( \frac{Q(c) - q_{min}}{q_{max} - q_{min}} + \lambda \cdot \sqrt{\frac{\ln(N(n_c) + 1)}{N(c)}} \right),$$
(5)

where $q_{max}$ and $q_{min}$ are the upper and lower limits of quality values $Q(\cdot)$ that ever encountered in MCTS, respectively. From the root $n_r$, MCTS iteratively selects a child node with the largest UCT value until reaching a leaf node.

*Expansion:* For the leaf node selected in the selection stage, the expansion stage prompts LLMs with actions e2, m1, m2, and s1 to build its child nodes. To attempt various detailed designs, in a single expansion stage, MCTS-AHD generates $k$ child nodes with the actions m1 and m2, one with e2 and s1, respectively ($2k + 2$ child nodes in total).

*Simulation:* Then, MCTS-AHD evaluates these newly generated heuristic functions $h$ on the evaluation dataset $D$ for their performances $g(h)$ and directly sets their quality value as their performances, i.e., $Q(n_l) \leftarrow g(h)$. We also record their visit count $N(n_l) \leftarrow 1$. Simultaneously, the elite set $E$ for the action e2, $q_{max}$, and $q_{min}$ will be updated.

*Backpropagation:* The backpropagation process updates the

quality values and the visit counts in MCTS as follows:

$$Q(n_c) \leftarrow \max_{c \in \text{Children}(n_c)} Q(c),$$
$$N(n_c) \leftarrow \sum_{c \in \text{Children}(n_c)} N(c).$$
(6)

**Progressive Widening**. Since the elite heuristic function set $E$ for the action e2 updates gradually as the search progresses. MCTS-AHD introduces the progressive widening technique to enable the re-exploration of non-leaf nodes, especially nodes with higher visit counts. The progressive widening process will occur when the condition in Eq.(4) is satisfied and we have $\alpha = 0.5$ for MCTS-AHD. We call the **action e1** for a new child node when the root node $n_r$ qualifies the condition in Eq.(4), and we call **action e2** when other nodes qualify. Heuristic functions inputted to the action e1 are uniformly selected from 2 to 5 different subtrees of MCTS. We implement the progressive widening process during the selection stage, then these nodes will also be processed in the simulation and backpropagation stages.

Eventually, MCTS-AHD will output a heuristic function with the highest performance value throughout the MCTS.

### 3.3. Exploration-Decay

The setting of the exploration factor $\lambda$ in Eq.(5) determines the preferences of MCTS on exploration or exploitation (Browne et al., 2012). A larger $\lambda$ promotes the exploration of temporarily inferior nodes, while a smaller $\lambda$ stimulates concentration on nodes with higher quality values $Q(\cdot)$. To facilitate a more comprehensive exploration in the early iterations of MCTS and ensure convergence in the later ones, MCTS-AHD presents an exploration-decay technique, linearly decaying the exploration factor $\lambda$ in Eq.(5) as follows:

$$\lambda = \lambda_0 * \frac{T - t}{T}.$$
(7)

Although having a task-specific setting $\lambda_0$ is helpful for high-quality heuristics, MCTS-AHD strives to keep the generalization ability, so we set $\lambda_0 = 0.1$ for all tasks.

## 4. Experiments

This section evaluates the proposed MCTS on designing outstanding heuristics for complex optimization tasks, including NP-hard CO problems and a Cost-aware Acquisition Function (CAF) design task for BO (Yao et al., 2024c). The definitions of these tasks are in Appendix B. We implement MCTS-AHD to design key functions of a wide range of general frameworks (detailed in Appendix C) for these tasks, including step-by-step construction, Ant Colony Optimization (ACO), Guided Local Search (GLS), and BO.

**Settings.** For experiments in this section, we set the number of initial tree nodes $N_I = 4$, $\lambda_0 = 0.1$, $\alpha = 0.5$. The

*Table 1.* Designing heuristics with the step-by-step construction framework for TSP and KP. We evaluate methods on 6 test sets with 1,000 instances each. Test sets with in-domain scales (i.i.d. to the evaluation dataset $D$) are underlined. Since AHD methods have no guarantees for generalization ability, the effect on in-domain datasets is more important. Optimal for TSP is obtained by LKH (Lin & Kernighan, 1973), and Optimal for KP is the result of OR-Tools. Each LLM-based AHD method is run three times and we report the average performances. The best-performing method with each LLM is shaded, and each test set's overall best result is in bold.

| Task | TSP | | | | | | KP | | | | | |
|---|---|---|---|---|---|---|---|---|---|---|---|---|
| N= | _N=50_ | | _N=100_ | | _N=200_ | | _N=50, W=12.5_ | | _N=100, W=25_ | | _N=200, W=25_ | |
| Methods | Obj.↓ | Gap | Obj.↓ | Gap | Obj.↓ | Gap | Obj.↑ | Gap | Obj.↑ | Gap | Obj.↑ | Gap |
| Optimal | 5.675 | - | 7.768 | - | 10.659 | - | 20.037 | - | 40.271 | - | 57.448 | - |
| Greedy Construct | 6.959 | 22.62% | 9.706 | 24.94% | 13.461 | 26.29% | 19.985 | 0.26% | 40.225 | 0.12% | 57.395 | 0.09% |
| POMO | **5.697** | **0.39%** | **8.001** | **3.01%** | 12.897 | 20.45% | 19.612 | 2.12% | 39.676 | 1.48% | 57.271 | 0.09% |
| LLM-based AHD: *GPT-3.5-turbo* | | | | | | | | | | | | |
| Funsearch | 6.683 | 17.75% | 9.240 | 18.95% | 12.808 | 19.61% | 19.985 | 0.26% | 40.225 | 0.12% | 57.395 | 0.09% |
| EoH | 6.390 | 12.59% | 8.930 | 14.96% | 12.538 | 17.63% | 19.994 | 0.21% | 40.231 | 0.10% | 57.400 | 0.08% |
| MCTS-AHD(Ours) | 6.346 | 11.82% | 8.861 | 14.08% | 12.418 | 16.51% | 19.997 | 0.20% | 40.233 | 0.09% | 57.393 | 0.10% |
| LLM-based AHD: *GPT-4o-mini* | | | | | | | | | | | | |
| Funsearch | 6.357 | 12.00% | 8.850 | 13.93% | 12.372 | 15.54% | 19.988 | 0.24% | 40.227 | 0.11% | 57.398 | 0.09% |
| EoH | 6.394 | 12.67% | 8.894 | 14.49% | 12.437 | 16.68% | 19.993 | 0.22% | 40.231 | 0.10% | 57.399 | 0.09% |
| MCTS-AHD(Ours) | 6.225 | 9.69% | 8.684 | 11.79% | **12.120** | **13.71%** | **20.015** | **0.11%** | **40.252** | **0.05%** | **57.423** | **0.04%** |

running time of each heuristic on the evaluation dataset $D$ is limited to 60 seconds. The composition of evaluation datasets $D$ for each task is detailed in Appendix D as well as the settings of general frameworks. Valuable LLM-based AHD methods should be flexible for various pre-trained LLMs, so we include both *GPT-3.5-turbo* and *GPT-4o-mini*.

**Baselines.** To verify the ability of heuristics designed by MCTS-AHD, we introduce four types of heuristics as baselines: **(a)** Manually designed heuristics, e.g., Nearest-greedy (Rosenkrantz et al., 1977), ACO (Dorigo et al., 2006), EI (Mockus, 1974). **(b)** Traditional AHD method: GHPP (Duflo et al., 2019) **(c)** Neural Combinatorial Optimization (NCO) methods under the same general frameworks, e.g. POMO (Kwon et al., 2020) and DeepACO (Ye et al., 2024b). **(d)** Existing LLM-based AHD methods: Funsearch (Romera-Paredes et al., 2024), EoH (Liu et al., 2024b), ReEvo (Ye et al., 2024a), and the most recent work HSEvo (Dat et al., 2024). Funsearch, ReEvo, and HSEvo design heuristics from a handcrafted low-quality seed function, and we provide the same seed function for each design scenario without providing external knowledge. Instead, running EoH and MCTS-AHD does not require manually setting a seed function to initiate the heuristic evolution, so both methods demonstrate better applicability.

We design heuristics with the proposed MCTS-AHD and LLM-based AHD baselines on a single Intel(R) i7-12700 CPU. Following similar settings of Liu et al. (2024b), for almost all tasks, we set the evaluation budget of LLM-based AHD methods on the evaluation dataset $D$ as $T = 1,000$. In designing heuristics for each application scenario, we conduct three independent runs for each LLM-based AHD method to reduce statistical biases. To verify the significant advantages of MCTS-AHD, we perform more runs on some tasks in Appendix F.4 to obtain the p-value. In designing heuristics with the step-by-step construction framework for

the 0-1 Knapsack Problem (KP), executing MCTS-AHD with $T=1,000$ takes approximately three hours, 1M input tokens, 0.2M output tokens, about 0.3$ with *GPT-4o-mini*. Compared to LLM-based AHD baselines, there is no significant efficiency degradation or cost improvement.

### 4.1. MCTS-AHD for NP-hard CO Problems

As commonly recognized complex tasks (Korte et al., 2011), this subsection evaluates MCTS-AHD on NP-hard CO problems, including TSP, KP, Capacitated Vehicle Routing Problem (CVRP), Multiple Knapsack Problem (MKP), Bin-Packing Problem (BPP) with both online and offline settings (online BPP & offline BPP), and Admissible Set Problem (ASP). For these NP-hard CO problems, we apply MCTS-AHD to automatically design heuristics with several general frameworks, including step-by-step construction, ACO, and GLS (GLS results are shown in Appendix F.2).

**Step-by-Step Construction Framework.** The step-by-step construction framework (also known as the constructive heuristic framework) is simple but flexible for task solving, which constructs nodes in feasible CO solutions one by one (Asani et al., 2023). Besides being a general framework for LLM-based AHD methods, it is also the most common framework adopted in NCO methods (Vinyals et al., 2015; Bello et al., 2016). We use MCTS-AHD to design heuristics with the step-by-step construction framework for TSP, KP, online BPP, and ASP (with ASP results in Appendix F.1).

*TSP & KP.* We first evaluate MCTS-AHD by designing TSP and KP heuristics within step-by-step construction frameworks. The key heuristic function to be designed should select the next TSP node or the next KP item to join taking the temporary solving state (e.g., selected and remaining TSP nodes or KP items) as inputs. This function will be executed recursively until a complete feasible solution is constructed.

*Table 2.* Designing heuristics with the ACO general framework for solving TSP, CVRP, MKP, and offline BPP. Each test set contains 64 instances and LLM-based AHD methods' performances are averaged over three runs.

| | TSP | | | | CVRP | | | | MKP | | | | Offline BPP | | | |
|---|---|---|---|---|---|---|---|---|---|---|---|---|---|---|---|---|
| Test sets | $N$=50 | | $N$=100 | | $N$=50, $C$=50 | | $N$=100, $C$=50 | | $N$=100, $m$=5 | | $N$=200, $m$=5 | | $N$=500, $C$=150 | | $N$=1,000, $C$=150 | |
| Methods | Obj.↓ | Gap | Obj.↓ | Gap | Obj.↓ | Gap | Obj.↓ | Gap | Obj.↑ | Gap | Obj.↑ | Gap | Obj.↓ | Gap | Obj.↓ | Gap |
| ACO | 5.992 | 3.28% | 8.948 | 9.40% | 11.355 | 27.77% | 18.778 | 25.76% | 22.738 | 2.31% | 40.672 | 4.30% | 208.828 | 2.81% | 417.938 | 3.15% |
| DeepACO | 5.842 | 0.71% | 8.282 | 1.26% | **8.888** | **0.00%** | **14.932** | **0.00%** | 23.093 | 0.79% | 41.988 | 1.20% | **203.125** | **0.00%** | **405.172** | **0.00%** |
| | LLM-based AHD: *GPT-4o-mini* | | | | | | | | | | | | | | | |
| EoH | 5.828 | 0.45% | 8.263 | 1.03% | 9.359 | 5.31% | 15.681 | 5.02% | 23.139 | 0.59% | 41.994 | 1.19% | 204.646 | 0.75% | 408.599 | 0.85% |
| ReEvo | 5.856 | 0.94% | 8.340 | 1.98% | 9.327 | 4.94% | 16.092 | 7.77% | 23.245 | 0.13% | 42.416 | 0.19% | 206.693 | 1.76% | 413.510 | 2.06% |
| HSEvo | 5.810 | 0.14% | 8.219 | 0.50% | 9.349 | 5.19% | 16.282 | 9.04% | **23.276** | **0.00%** | 42.494 | 0.01% | 205.563 | 1.20% | 410.745 | 1.38% |
| MCTS-AHD(Ours) | **5.801** | **0.00%** | **8.179** | **0.00%** | 9.286 | 4.48% | 15.782 | 5.70% | 23.269 | 0.03% | **42.498** | **0.00%** | 204.094 | 0.48% | 407.323 | 0.53% |

For all LLM-based AHD methods, the evaluation dataset $D$ contains 64 50-node ($N$=50) TSP instances for TSP and 64 100-item ($N$=100) KP instances with capacity $W = 25$ for KP. Table 1 shows the performance of baseline heuristics and LLM-based AHD methods. The Greedy Construct baseline is the Nearest-greedy heuristic algorithm for TSP and it constructs KP solutions by the item with the largest value-weight-ratio. MCTS-AHD exhibits significant advantages on almost all test sets, surpassing manually designed heuristics and LLM-based AHD methods EoH and Funsearch. Moreover, compared to an advanced NCO method POMO which requires task-specific training, MCTS-AHD can design better heuristics in 200-node TSP and KP test sets, exhibiting its power to solve NP-hard CO problems.

*Online BPP*. As another widely considered NP-hard CO problem, online BPP is the online variant of BPP. It allows only immediate bin packing decisions once a new item is received. It is generally adopted as a common evaluation scenario for LLM-based AHD methods. We follow (Liu et al., 2024b) to generate WeiBull BPP instances (Castiñeiras et al., 2012) and use four WeiBull instances with diverse scales as the evaluation dataset $D$. As shown in Table 3, online BPP heuristics designed by MCTS-AHD demonstrate a superior average performance of six test sets.

*Table 3.* Design step-by-step construction heuristics for solving online BPP. The table exhibits the performance gaps of heuristics to the lower bound. Each LLM-based AHD method is run three times for the average gaps. Each test set contains five WeiBull BPP instances and we underline the four (in-domain) scales contained in $D$. The scale notations of test sets are abbreviated, e.g., 1k_100 represents 1,000 items and capacity $W = 100$.

| Online BPP | | | | | | |
|---|---|---|---|---|---|---|
| Test sets | 1k_100 | 1k_500 | 5k_100 | 5k_500 | 10k_100 | 10k_500 | Avg. |
| Best Fit | 4.77% | 0.25% | 4.31% | 0.55% | 4.05% | 0.47% | 2.40% |
| First Fit | 5.02% | 0.25% | 4.65% | 0.55% | 4.36% | 0.50% | 2.56% |
| | LLM-based AHD: *GPT-4o-mini* | | | | | | |
| Funsearch | 2.45% | 0.66% | 1.30% | 0.25% | 1.05% | 0.21% | 0.99% |
| EoH | 2.69% | 0.25% | 1.63% | 0.53% | 1.47% | 0.45% | 1.17% |
| ReEvo | 3.94% | 0.50% | 2.72% | 0.40% | 2.39% | 0.31% | 1.71% |
| HSEvo | 2.64% | 1.07% | 1.43% | 0.32% | 1.13% | 0.21% | 1.13% |
| MCTS-AHD | 2.45% | 0.50% | 1.06% | 0.32% | 0.74% | 0.26% | **0.89%** |

**Ant Colony Optimization Framework**. The ACO is an optimization algorithm inspired by the foraging behavior of ants, which contains a heuristic matrix and implements a path selection mechanism to solve combinatorial optimization problems by simulating the transfer of pheromones between ants (Dorigo et al., 2006; Kim et al., 2024). LLM-based AHD can design a generation function of the heuristic matrix, thereby transforming ACO into a framework and applying it to a variety of tasks. Following Ye et al. (2024a), MCTS-AHD designs heuristics within the ACO framework for four NP-hard CO problems: TSP, CVRP, MKP, and offline BPP. The design results using *GPT-4o-mini* as LLMs are shown in Table 2, where MCTS-AHD exhibits significant leads to EoH and ReEvo in three in-domain test sets across four CO problems and three out-of-domain test sets. Moreover, the proposed MCTS-AHD can consistently outperform manually designed ACO heuristics in all eight test sets and surpass an outstanding NCO method DeepACO (Ye et al., 2024b) in TSP and MKP test sets.

### 4.2. MCTS-AHD for Other Complex Tasks

To assess whether MCTS-AHD can still perform well in optimization tasks beyond CO problems, we follow Yao et al. (2024c) to evaluate MCTS-AHD by designing heuristic CAFs for BO. The CAF is a crucial component for Cost-aware BO, helping to reach the global optimum in a cost-efficient manner. There are several advanced hand-crafted heuristic CAFs, including EI (Mockus, 1974), EIpu (Snoek et al., 2012), and EI-cools (Lee et al., 2020a). We employ two synthetic instances with different landscapes and input dimensions (Ackley and Rastrigin in Table 4) as the evaluation dataset $D$ for LLM-based AHD and also test the manually and automatically designed heuristics on ten other synthetic instances. During heuristic evolutions, we set the sampling budget to 12 and run 5 independent trials for average performances. As shown in Table 4, heuristic CAFs designed by the proposed MCTS-AHD demonstrate superior BO results that outperform both manually designed heuristics and EoH in six out of twelve synthetic instances. It verifies that MCTS-AHD not only performs well in NP-

*Table 4.* Designing CAFs for BO. The table shows the gaps to optimal when running BO on instances with manually designed CAFs and CAFs designed by LLM-based AHD methods. LLM-based AHD methods are run three times for the average gaps. In testing, the evaluation budgets for BO are 30 and we run 10 trials for average gaps. The results of EI, EIpu, and EI-cool are from Yao et al. (2024c).

| Instances | Ackley | Rastrigin | Griewank | Rosenbrock | Levy | ThreeHumpCamel | StyblinskiTang | Hartmann | Powell | Shekel | Hartmann | Cosine8 |
|---|---|---|---|---|---|---|---|---|---|---|---|---|
| EI | 2.66% | 4.74% | 0.49% | **1.26%** | **0.01%** | 0.05% | 0.03% | 0.00% | 18.89% | 7.91% | 0.03% | 0.47% |
| EIpu | 2.33% | 5.62% | 0.34% | 2.36% | 0.01% | 0.12% | **0.02%** | 0.00% | 19.83% | 7.92% | 0.03% | 0.47% |
| EI-cool | 2.74% | 5.78% | **0.34%** | 2.29% | 0.01% | 0.07% | 0.03% | **0.00%** | 14.95% | 8.21% | **0.03%** | 0.54% |
| *LLM-based AHD: GPT-4o-mini* | | | | | | | | | | | | |
| EoH | 2.45% | 0.90% | 0.54% | 56.78% | 0.20% | 0.26% | 0.79% | 0.04% | 70.89% | 4.56% | 0.33% | 0.36% |
| MCTS-AHD | **2.40%** | **0.77%** | 0.36% | 1.68% | 0.01% | **0.02%** | 0.20% | 0.01% | **1.27%** | **3.94%** | 0.38% | **0.34%** |

hard CO problems but can also show great power in other complex optimization tasks.

# 5. Discussion

Experiments have demonstrated the effectiveness of the proposed MCTS-AHD in designing high-quality heuristics for a wide range of application scenarios. This section first conducts essential ablation studies. Then, we will also analyze the advantages of utilizing MCTS in LLM-based AHD compared to the original population-based EC.

## 5.1. Ablation on Parameters and Components

To validate the necessity of its components, as shown in Table 5, we first remove three proposed components of MCTS-AHD (Progressive Widening, Thought-alignment, and Exploration-decay) and use these variants to design step-by-step heuristics for TSP and KP. Line 3 to Line 5 in Table 5 reports their average optimality gaps over three runs, and all these variants exhibit a clear performance degradation in at least one task compared to the original MCTS-AHD.

The actions for node expansion in MCTS-AHD are also essential. Actions e1 and e2 are associated with the progressive widening, so they cannot be ablated individually. According to Table 5, MCTS-AHD variants without actions s1, m1, and m2 could only design worse heuristics in at least one task, proving the significance of these LLM-based

*Table 5.* Ablations on the components, actions, and parameter settings of MCTS-AHD. We use MCTS-AHD variants to design heuristics with the step-by-step construction framework for their optimality gaps on 1,000-instance test sets averaged over **five runs**.

| Methods | TSP50 | KP100 |
|---|---|---|
| MCTS-AHD (Original, 10 runs) | 10.661% | 0.059% |
| *w/o* Progressive Widening | 12.132% | 0.064% |
| *w/o* Thought-alignment | 11.640% | 0.061% |
| *w/o* Exploration-decay | 11.606% | 0.064% |
| *w/o* Action s1 | 11.919% | 0.062% |
| *w/o* Action m1 | 10.921% | 0.083% |
| *w/o* Action m2 | 11.679% | 0.061% |
| MCTS-AHD variant $\lambda_0 = 0.05$ | 11.080% | 0.056% |
| MCTS-AHD variant $\lambda_0 = 0.2$ | 12.124% | 0.034% |

actions. The importance of action s1 also demonstrates that MCTS-AHD benefits from its organized tree structure.

Meanwhile, we measure the sensitivity of the main parameter $\lambda_0$. Results in the last two lines of Table 5 show that although the TSP and KP tasks may have respective preferences in the $\lambda_0$ setting, the default setting (i.e., $\lambda_0 = 0.1$) exhibits generally good quality.

## 5.2. MCTS versus Population-based EC

**Ability of MCTS-AHD in Escaping from Local Optima**. As the main contribution of this paper, instead of population-based EC, MCTS-AHD can manage inferior but potential heuristic functions, achieving a more comprehensive exploration of the heuristic space $H$ thus avoiding falling into local optima. To verify this, we plot the performance curves of MCTS-AHD and LLM-based AHD baselines in designing step-by-step construction heuristics for TSP and designing CAFs for BOs. Each curve is averaged over at least **five runs**. As illustrated in Figure 4, all baseline methods with populations exhibit early convergences on performances, but MCTS-AHD can converge to significantly better performance via quick and continuous performance updates.

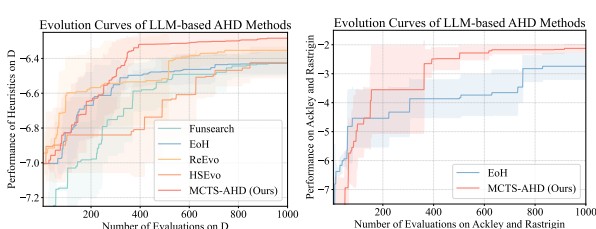

(a) Design Step-by-step Construction heuristics for TSP

(b) Design CAFs for BO

*Figure 4.* Evolution curves on two diverse application scenarios.

**Advantage scopes of MCTS-AHD**. Compared to population-based baselines, we claim that MCTS-AHD demonstrates greater advantages in application scenarios with more complex heuristic spaces $H$ and application scenarios with more descriptions as knowledge. We analyze these two claims with experiments in Appendix F.10.

## 6. Conclusion

In conclusion, this paper first applies MCTS to LLM-based AHD. The proposed MCTS-AHD achieves a comprehensive exploration of the heuristic space and can finally design higher-quality heuristics for NP-hard complex tasks. For LLM-based AHD, MCTS can be a more promising evolution method compared to population-based EC.

**Limitation and Future Work**: As a limitation of MCTS-AHD, the convergence speed of MCTS-AHD can still be improved. In the future, we will consider designing MCTS-population hybrid methods for better evolution efficiency.

## Impact Statement

This paper presents work whose goal is to advance the field of Machine Learning. There are many potential societal consequences of our work, none which we feel must be specifically highlighted here.

## Acknowledge

This work was partially supported by the National Natural Science Foundation of China (Grant No. 62476118), the Natural Science Foundation of Guangdong Province (Grant No. 2024A1515011759), the Natural Science Foundation of Shenzhen (Grant No.JCYJ20220530113013031) and the Guangdong Science and Technology Program (Grant No. 2024B1212010002).

Moreover, we would like to express our esteemed gratitude to Professor Wee Sun Lee for his valuable suggestions on this work.

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

# A. Related Work

This section presents a detailed literature review of various research areas related to LLM, AHD, MCTS, and CO problems.

## A.1. AHD

Automatic Heuristic Design, also known as Hyper-Heuristics, (Burke et al., 2013) aims to find the best-performing heuristic among an extensive set of valid heuristics, i.e., the heuristic space. As the search strategy, AHD generally adopts the evolutionary algorithm to update heuristic algorithms automatically (Stützle & López-Ibáñez, 2019) using various effective methodologies and frameworks (Blot et al., 2016; López-Ibáñez et al., 2016; Burke et al., 2019). GP (Mei et al., 2022) is a prevailing and effective approach for AHD. However, it requires hand-crafted operators for heuristic mutation and crossover (Duflo et al., 2019; Sánchez-Díaz et al., 2021).

## A.2. Neural Combinatorial Optimization (NCO)

Traditional methods for NP-hard CO problems contain exact algorithms (Fischetti et al., 2007; Lusby et al., 2010) and approximation algorithms (including handcrafted heuristics) (Merz & Freisleben, 1997; Helsgaun, 2017). Integrating the deep learning technique (He et al., 2016; Vaswani et al., 2017), Neural Combinatorial Optimization (NCO) methods can obtain near-optimal results with less time consumption compared to traditional methods (Bello et al., 2016; Kool et al., 2018). These methods train neural networks for decision-making with diverse solving process (e.g., the LKH algorithm (Xin et al., 2021), the GLS algorithm (Sui et al., 2024; Hudson et al., 2021), improvement-based framework (Wu et al., 2021; Zheng et al., 2023), or step-by-step construction (Vinyals et al., 2015; Zheng et al., 2024a; Luo et al., 2025a)). NCO methods can be regarded as a special kind of Hyper-Heuristics (AHD) (Ye et al., 2024a), where the solving process is the general framework of AHD and training NCO methods searches for the optimal parameter settings within a parameterized heuristic space. In contrast to traditional heuristics, NCO does not require expert knowledge so some NCO methods can be applied to multiple CO problems (Ye et al., 2024c; Zheng et al., 2024b; Luo et al., 2025b; Liu et al., 2024a; Drakulic et al., 2024; Berto et al., 2024).

Compared to NCO methods, designing heuristics with LLM-based AHD methods for NP-hard CO problems demonstrates better efficiency, better applicability, and lower implementation difficulty. Training an NCO model on a given framework would take several days or even several weeks on an advanced GPU (Kwon et al., 2020), whereas a high-quality heuristic can be generated in a few hours using the LLM-based AHD methods without GPU requirements. Considering applicability, NCO methods require special framework and network designs to solve special CO problems (Kwon et al., 2021), but LLM-based AHD methods can be applied to new CO problems without task-specific adaptations. NCO methods also bring implementation difficulties when creating complicated task environments for model training (Zheng et al., 2024b). Instead, LLM-based AHD methods only need some linguistic descriptions as the task environment. Moreover, LLM-based AHD methods can even demonstrate better results than NCO methods in some application scenarios, as shown in Table 1, Table 2, and Table 8, heuristics designed by the proposed MCTS-AHD can outperform advanced NCO methods under the same solving framework (e.g. POMO (Kwon et al., 2020), DeepACO (Ye et al., 2024b)) in 10 out of 16 test sets.

## A.3. LLM for EC

EC is a generic optimization principle inspired by natural evolution (Bäck et al., 1997; Eiben & Smith, 2015). The core idea of EC is to generate a set of candidate solutions (called populations) by simulating genetic variation and natural selection in the process of biological evolution and to gradually optimize these solutions through an iterative process over multiple generations. Some recent work focuses on stimulating the crossover or mutation operators in the original evolutionary computation framework by prompting LLMs (Lehman et al., 2023; Meyerson et al., 2023; Lange et al., 2024) or introducing LLMs to generate auxiliary information (Ye et al., 2024a). LLM-based EC approaches can be specified to plenty of application scenarios, including planning (Kambhampati et al., 2024), code generation (Hemberg et al., 2024), and LLM-based AHD discussed in this article (Romera-Paredes et al., 2024).

## A.4. LLM for AHD

LLM-based AHD methods with populations can be summarized as a four-part process (Liu et al., 2023a; Romera-Paredes et al., 2024; Liu et al., 2024b; Ye et al., 2024a; Dat et al., 2024; Liu et al., 2024d), **1):** Heuristics Initialization: LLM-based AHD methods employ LLMs to generate multiple initial heuristic functions or directly handcraft seed functions as initial

heuristic functions. **2):** Operator selection: Existing methods design a variety of prompts corresponding to mutation or crossover operators on parent heuristics. In this step, population-based methods will randomly select an operator to be executed. Afterward, these methods also select the parent heuristic(s) to be operated from the current population (Yin et al., 2024). **3):** New heuristic generation: Next, LLM-based AHD methods will prompt LLMs for the Python code implementation of a new heuristic function based on the description of the task $P$, general framework, the prompt of selected operators, and the selected parent heuristics. **4):** Population update: As the final step in each iteration, each newly generated heuristic $h$ will run on the evaluation dataset $D$ to obtain $g(h)$. The invalid heuristics will be discarded. The population will update according to performances $g(\cdot)$ of heuristics. The last three processes will run in a loop until the number of heuristic evaluations reaches a given budget $T$.

Yao et al. (2024a) considers multiple objectives (for example, heuristic performance and execution efficiency) in LLM-based AHD. This article focuses only on heuristic performances, so we do not involve it as a baseline.

### A.5. LLM for CO

Besides LLM-based AHD, there are also two ways to utilize LLMs for CO problems, including LLM-as-optimizer methods and (fine-tuned) LLM-as-solver methods. Yang et al. (2024) first develop LLMs as the optimizer for TSP tours. LLM-as-optimizer methods (Guo et al., 2023; Liu et al., 2024e) initially generate several feasible solutions for a single CO instance. Then, the LLM is iteratively prompted for better solutions by taking the existing top-performing solutions and their objective values as in-context information. Then, the newly generated solution in each iteration is evaluated for its objective value and inserted back into the in-context. LLM-as-solver methods directly treat LLMs as end-to-end CO instance solvers (Abgaryan et al., 2024; Jiang et al., 2024). These methods consider LLMs as pre-trained NCO models and establish environments to fine-tune LLMs for better performances on CO instances.

LLM-based AHD methods, especially MCTS-AHD, are still the most promising directions for solving complex NP-hard CO problems. LLM-as-optimizer methods pose high demand for LLM's reasoning ability and knowledge storage, and current LLMs (Huang et al., 2024) can only offer extremely limited results on infamous and large-scale CO instances (refer to comparisons in Table 14). It should be noted that Kambhampati et al. (2024) also recommends using LLMs for module design instead of instance solving for planning tasks, where the module design process corresponds to the idea of LLM-based AHD. (Fine-tuned) LLM-as-solver methods (Abgaryan et al., 2024; Jiang et al., 2024) require additional training and face difficulties in configuring training environments. Moreover, unlike NCO models that represent each coordinate value as a piece of embeddings, LLMs face challenges in tokenizing high-precision coordinate numbers in a linguistic way (Wu et al., 2024).

### A.6. LLM Inference with MCTS

With the great success of CoT (Wei et al., 2022) and ToT (Yao et al., 2024b) in enhancing the ability of LLM reasoning, a series of System 2 techniques (Weston & Sukhbaatar, 2023) have been proposed in various applications (Wang et al., 2025; Zheng & Lee, 2025). Among them, MCTS has recently emerged as a powerful technique to enhance the reasoning capabilities of LLMs (Zhang et al., 2024b). These methods mainly construct MCTS in two ways (Feng et al., 2023), representing each node with a complete answer refined from the parents (Zhang et al., 2024a; Zhou et al., 2023) or a reasoning step following its parents (Qi et al., 2024). The MCTS-AHD in this paper belongs to the former, where each node represents a complete piece of executable function and its description. When dealing with commonsense QA or mathematical problems, System 2 techniques often use self-evaluation for MCTS simulation (Zhang et al., 2024a), but MCTS-AHD does not involve the self-evaluation or rollout (Silver et al., 2016) since AHD methods can easily obtain performances $g(\cdot)$ for tasks.

### A.7. LLM for Code Generation

Recent LLMs have strong coding capabilities (Zhang et al., 2024c), and some recent papers have designed a series of structures based on reflection and MCTS (Dainese et al., 2024; DeLorenzo et al., 2024) to promote this ability. The LLM for code generation is a similar domain to LLM-based AHD. Compared to code generation tasks aiming at passing algorithm tests, LLM-based AHD faces significantly more challenges in finding the optimal heuristic (Liu et al., 2024c). LLM-based AHD needs to consider more on exploring the entire heuristic space and optimizing code performance by modifying parameter settings and detail designs.

As a similar method to this paper, Brandfonbrener et al. (2024) uses MCTS with the progressive widening technique for code generation, but its specific MCTS and progressive widening procedure are totally different from the proposed MCTS-AHD.

### A.8. Connection to General LLM Applications

Nowadays, language models are dominant in a wide range of AI applications, and there are increasing concerns about their interpretability (Liu et al., 2025; 2024g), causal reasoning (Liu et al., 2023b; 2024f), and robustness (Liu et al., 2023c; 2022). We acknowledge its growing importance and will consider it in the future of our research.

## B. Definition of Tasks

### B.1. NP-hard CO Problems

This paper conducts experiments on six NP-hard representative CO problems, including TSP, CVRP, KP, MKP, ASP, and BPP. For BPP, we consider both its online setting (online BPP) and offline setting (offline BPP). In this section, we will introduce the mathematical definitions of these CO problems in detail.

**Traveling Salesman Problem**  TSP is one of the most representative COPs (Biggs, 1986), which aims at finding the shortest path to visit each city once and returns to the starting point. An $N$-node TSP instance $\boldsymbol{ins}$ contains distance matrix $\{\boldsymbol{D} = d_{j,k}, j = 1, \ldots, N, k = 1 \ldots, N\} \in \mathbb{R}^{N \times N}$, where $d_{j,k}$ denotes the cost between nodes $k$ and $j$, the goal is to minimize the following objective function (Zheng et al., 2024b):

$$\text{minimize} \quad f(\boldsymbol{s}) = \sum_{t=1}^{N-1} d_{s_t, s_{t+1}} + d_{s_N, s_1}, \tag{8}$$

where the solution $\boldsymbol{s} = (s_1, s_2, \ldots, s_N)$ is a permutation of all node indexes. All the feasible solutions satisfy the constraint of the degree of each node being two and containing no loop with lengths less than N.

**Capacitated Vehicle Routing Problem**  CVRP aims to plan several capacity-constrained vehicles starting at and returning to a depot, meeting the demands of multiple customers, and minimizing the total travel distance. Each CVRP instance contains a depot (the 0-th node) and several customers. With a distance matrix $\{\boldsymbol{D} = d_{j,k}, j = 0, \ldots, N, k = 0 \ldots, N\}$, the CVRP can be expressed as follows:

$$\text{minimize} \quad f(\boldsymbol{s}) = \sum_{j=1}^{q} C(\boldsymbol{\rho}^j), \quad C(\boldsymbol{\rho}^j) = \sum_{t=0}^{|\boldsymbol{\rho}^j|-1} d_{\rho_t^j, \rho_{t+1}^j} + d_{\rho_{n_j}^j, \rho_0^j},$$

$$\text{s.t.} \quad 0 \le \delta_i \le C, \quad \sum_{i \in \boldsymbol{\rho}^j} \delta_i \le C, \quad i \in \{1, \ldots, n\}, j \in \{1, \ldots, q\}, \tag{9}$$

where $\boldsymbol{s}$ is a solution representing the complete route of vehicles and consists of $q$ sub-routes $\boldsymbol{s} = \{\boldsymbol{\rho}^1, \boldsymbol{\rho}^2, \ldots, \boldsymbol{\rho}^q\}$. Each sub-route $\boldsymbol{\rho}^j = (\rho_1^j, \ldots, \rho_{n_j}^j)$, $j \in \{1, \ldots, q\}$ starts from the depot $s_0$ and goes back to $s_0$, $n_j$ represents the number of customer nodes in it. $n = \sum_{j=1}^q n_j$ is the total number of customer nodes; $\delta_i$ denotes the demand of node $i$; $C$ denotes the capacity of the vehicle. This paper follows the settings of ReEvo (Ye et al., 2024a) when generating CVRP data sets, fixing the depot coordinates to (0.5, 0.5).

**0-1 Knapsack Problem**  KP is a typical CO problem, consider loading items of a maximum total value to a $W$-capacity knapsack. Each item can only be picked once. KP solution $\boldsymbol{s} \subseteq \{1, 2, ..., N\}$ records the selection item indexes. A KP instance $\boldsymbol{ins}$ records the value $v_i \sim \text{Uniform}(0, 1)$ and weight $w_i \sim \text{Uniform}(0, 1)$ of each candidate item. We follow the settings in Kool et al. (2018) in generating instances and have $W = 25$ for 100-item and 200-item KP instances and $W = 12.5$ for 50-item ones.

We adopt a traditional greedy heuristic algorithm for the Greedy Construct in Table 1 (Kwon et al., 2020). This algorithm starts with an empty knapsack (as solution $\boldsymbol{s}$) and recursively adds the item that meets the capacity limit and has the maximum value-weight-ratio ($\frac{v_i}{w_i}$) from the remaining items into the current knapsack (add this item to solution $\boldsymbol{s}$ as well). The algorithm will stop when the knapsack can no longer load more items.

**Multiple Knapsack Problem**    We follow the ReEvo (Ye et al., 2024a) settings for MKP instances, uniformly sampling the value $v_i \sim \text{Uniform}(0, 1)$ and weight for the $m$ knapsack $w_{ij} \sim \text{Uniform}(0, 1), i \in \{1, \ldots, m\}, j \in \{1, \ldots, n\}$. We also uniformly sample the capacity of the $m$ knapsacks $C_i, i \in \{1, \ldots, m\}$ from $(\max_j w_{ij}, \sum_j w_{ij})$.

**Admissible Set Problem**    ASP constructs a set of $n$-dimensional vectors $\mathcal{A}$ (called an admissible set) where vectors $\boldsymbol{a} \in \mathcal{A} \subset \{0, 1, 2\}^n$ and the number of non-zero items is constrained to be $w$. ASP aims to maximize the size of the admissible set $\mathcal{A}$ under a certain constraint that for any three distinct vectors there is a coordinate in which their three respective values are $\{0, 0, 1\}, \{0, 0, 2\}$, or $\{0, 1, 2\}$. We formulate the objective function and constraints as follows:

$$
\begin{aligned}
\text{maximize} \quad & |\mathcal{A}| \\
\text{s.t.} \quad & \sum_{i=1}^{n} \mathbb{I}[a_i \neq 0] = w, \exists i \in \{1, \ldots, n\}, \{a_i, b_i, c_i\} \in \{\{0, 0, 1\}, \{0, 0, 2\}, \{0, 1, 2\}\}, \forall \boldsymbol{a}, \boldsymbol{b}, \boldsymbol{c} \in \mathcal{A},
\end{aligned}
\tag{10}
$$

where $\mathbb{I}[a_i \neq 0]$ represents the number of non-zero items in vector $\boldsymbol{a} = (a_1, \ldots, a_n)$.

**Bin Packing Problem**    BPP is a classic NP-hard CO problem that aims to place a set of items with different sizes into as few $W$-capacity bins as possible. Online BPP requires making an immediate decision on which bin to place once a new item is received, while offline BPP does not require this. We follow Romera-Paredes et al. (2024) and Liu et al. (2024b) to generate WeiBull instances for online BPP and follow Ye et al. (2024a) to generate offline BPP instances with $W = 150$ and the size of items uniformly sampled from 20 to 100.

### B.2. Cost-Aware Acquisition Function Design in Bayesian Optimization

Please refer to Appendix C.4 for relevant introductions.

# C. Definition of General Frameworks

For NP-hard combinatorial optimization (CO) problems, we employ MCTS-AHD to design key functions within given general frameworks. To verify the framework-agnosticism of MCTS-AHD, we include three general frameworks for CO in experiments, e.g., step-by-step construction, GLS, and ACO, as well as the CAF design task using BO as the outer framework. Next, in this section, we will introduce these involved general frameworks in detail.

### C.1. Step-by-Step Construction

Step-by-step construction is an intuitive framework capable of handling almost all CO problems. It considers gradually extending a solution ($s$) of an NP-hard CO problem from scratch until a complete and feasible solution is constructed. In each step of the construction (i.e., solution extension), the step-by-step construction framework assigns priority to each candidate (decision variable), and the candidate with the highest priority will be added to the solution.

In the step-by-step construction framework, MCTS-AHD and LLM-based AHD baselines design the same key heuristic function that will be repeatedly executed to calculate the priority of candidates. This paper adopts the step-by-step construction framework for solving four CO problems, including TSP, KP, ASP, and online BPP. Built on this framework, Nearest-greedy is a prevailing manually designed heuristic for TSP that scores candidate nodes based sorely on their distance from the current node. Similarly, a greedy process that evaluates KP items based on their value-weight ratio is also commonly used. We consider these two handcrafted heuristics as the Greedy Construct baseline in Table 1. Additionally, in both TSP and KP, there is a series of NCO methods that train deep neural networks based on the step-by-step construction framework. Their neural networks can be considered as more efficient and sophisticated key functions for evaluating all candidate nodes or items. For involved CO problems, the detailed settings of the key heuristic function in the step-by-step construction frameworks are as follows:

- For TSP, MCTS-AHD is responsible for designing a function to select the next node to visit based on node coordinates, starting point, distance matrix, and all unvisited nodes.

- For KP, MCTS-AHD is responsible for selecting the next item to add to the knapsack based on the value and weight of all items to be chosen and the remaining capacity of the knapsack.

- For ASP, the key function provides a score for the current vector to determine to what extent it is suitable to be added to the admissible set ($\mathcal{A}$).

- In MCTS-AHD, the function required for online BPP gives a preference score for adding the current newly input item to each bin based on the item's size and the remaining capacity of all bins.

### C.2. Ant Colony Optimization

ACO is a meta-heuristic and evolutionary algorithm (EA) inspired by the behavior of ants to find the shortest route between their colony and food sources (Dorigo et al., 2006; Ansari & Daxini, 2022).

ACO records a pheromone matrix $\boldsymbol{\tau}$ and a heuristic matrix $\boldsymbol{\eta}$. Each item in the matrix $\tau_{ij}$ indicates the priority of including that edge $(i, j)$ in a solution. The pheromone trails are iteratively updated based on the quality of the solutions found, encouraging future ants to follow better paths. The heuristic information on each edge (i.e., $\eta_{ij}$) is a problem-specific measure that indicates the immediate benefit of choosing a particular path. For solving TSP with ACO and a manually designed heuristic matrix, $\eta_{ij}$ is often set to the inverse of the distance between cities $i$ and $j$, that is, $\eta_{ij} = \frac{1}{d_{ij}}$. LLM-based AHD methods are employed to design a more effective heuristic matrix $\boldsymbol{\eta}$ based on the necessary problem-specific inputs.

Ants construct solutions by moving from node to node, probabilistically choosing the next node based on a combination of pheromone and heuristic information. After all the ants have constructed their solutions, the pheromone levels update. An ACO iteration typically involves solution construction, optional local search, and pheromone update. By iteratively applying these steps, ACO algorithms can effectively explore the solution space and converge toward optimal or near-optimal solutions for NP-hard CO problems. We follow the settings in Ye et al. (2024a), evaluating MCTS-AHD by designing the heuristic metrics generation functions for TSP, CVRP, MKP, and offline BPP as follows:

- For TSP, the function inputs the distance matrix. We adopt the parameter settings in ReEvo (Ye et al., 2024a), the number of ants is set to 30 during heuristic evaluation (when evaluating on the evaluation dataset $D$), and the number

of iterations is 100. In testing, we allow more optimization for all ACO baselines and increase the number of iterations to 500.

- Functions for CVRP input the distance matrix, coordinates and demands of nodes, and the capacity $C$. Numbers of ants and iterations are set the same as TSP.

- For MKP, the function inputs the values and weights of items. The number of ants is set to 10. The number of iterations is 50 for running on the evaluation dataset $D$ and 100 for test sets.

- For offline BPP, the function inputs the size of items and the capacity of bins. The number of ants is set to 20. The number of iterations is 15 for running on the evaluation dataset $D$ and 100 for test sets.

**ACO for Black-box Settings**    Black-box settings are proposed in ReEvo (Ye et al., 2024a) as novel application scenarios for LLM-based AHD. The black-box settings do not provide descriptions for the task $P$ and the predefined general framework for heuristic design. The names of inputs in the to-be-designed key function will also be erased. These settings can simulate the application scenarios that cannot find any linguistic descriptions. This paper will evaluate AHD methods on both regular settings and black-box settings in Appendix F.10. For a better initial perception of the heuristic space $H$, we set $N_I = 10$ for MCTS-AHD in designing heuristics with black-box settings.

### C.3. Guided Local Search

The GLS framework uses local search algorithms (e.g., using 2-opt operators) for solution iterations and introduces a penalty mechanism to guide the search process escaping local optima. It shows capability on a wide range of CO problems with manually designed key functions (Voudouris & Tsang, 1999; Alhindi et al., 2019). We use LLM-based AHD methods to design the key heuristic function for generating knowledge-based matrices in the knowledge-guided local search (KGLS) framework (Arnold & Sörensen, 2019). Taking the TSP as an example, KGLS maintains a TSP solution while also preserving the ever-encountered best-performing solution. In each iteration of KGLS, KGLS first perturbs the TSP solution based on the generated knowledge matrix and then performs local searches using both 2-opt and relocate operators (Sengupta et al., 2019; Tuononen, 2022). We conduct 1200 iterations when running GLS heuristics on both the heuristic evolution process and testing for each CO instance. The number of perturbations to each solution is set to 30.

### C.4. Bayesian Optimization

BO (Shahriari et al., 2015) is a method for solving the black-box optimization problem where the objective is to find the global minimum of an unknown function $f(\mathbf{x})$ over a search space $\mathcal{X}$, represented as:

$$\mathbf{x}^* = \arg\min_{\mathbf{x} \in \mathcal{X}} f(\mathbf{x}). \tag{11}$$

Two core components of BO are the probabilistic surrogate model and the acquisition function (Mockus, 1974; Lam et al., 2016). In each iteration of BO, the probabilistic surrogate model is first trained using the available samples in the search space (BO typically employs a Gaussian process (GP) model (Williams & Rasmussen, 2006)). Then, the acquisition function utilizes the posterior information of the surrogate model to guide the subsequent search. Specifically, the next solution to evaluate at iteration $t$, denoted as $\mathbf{x}_t$, is chosen by maximizing the acquisition function:

$$\mathbf{x}_t = \arg\max_{\mathbf{x} \in \mathcal{X}} \alpha(\mathbf{x}, M_t), \tag{12}$$

where $M_t$ represents the information from the surrogate model at iteration $t$, and $\alpha(\mathbf{x}, M_t)$ represents the acquisition function value at $\mathbf{x}$. After performing an expensive evaluation at $\mathbf{x}_t$, this point is added to the available training dataset of samples. BO iteratively executes surrogate benchmark training and sampling based on the acquisition function until a termination criterion is met. Finally, the best solution among those evaluated samples is returned as the solution to the problem.

This article focuses on the cost-aware BO (Yao et al., 2024c; Luong et al., 2021; Snoek et al., 2012), which takes the number of expensive sample evaluations as the termination criterion. It not only focuses on optimizing the objective function but also considers the cost of evaluating the objective function. This method is beneficial in practical applications, especially when the evaluation cost of the objective function is high, like experimental design and hyper-parameter optimization of

machine learning models. BO employs a CAF as the acquisition function in cost-aware settings to manage the sample efficiency, which is the design focus of the cost-aware BO methods. We adopt LLM-based AHD methods for designing the CAF. During CAF evolutions, MCTS-AHD and EoH run 5 independent cost-aware BO trails with at most 12 function samples. In testing, we set the evaluation budget to 30 and report the average result of 10 trials.

## D. Details of Evaluations & Experiments

In this section, we provide a more detailed introduction to the setup of evaluation budgets $T$ and evaluation datasets $D$ used in the heuristic evaluation phase. Evaluation Settings in this paper are generally adopted from Funsearch (Romera-Paredes et al., 2024), EoH (Liu et al., 2024b), and ReEvo (Ye et al., 2024a).

**The Setting of $T$.** The number of generations in EoH is set to 20. Its population size is 20 for online BPP and 10 for TSP and FSSP (810 evaluations for TSP, FFSP, and 1620 for online BPP). So, this paper designs a similar number for the maximum number of evaluations $T$, setting the $T$ to 2,000 for online BPP (under the step-by-step construction framework) and setting $T = 1,000$ for other tasks.

**The Setting of $D$.** For most involved tasks, MCTS-AHD adopts the same settings in LLM-based baseline methods (e.g., EoH, ReEvo, Funsearch) for the evaluation dataset $D$. As a special case, for online BPP under the step-by-step construction framework, baselines adopt 5 5,000-item WeiBull BPP instances with $W$=100. However, such a dataset setting often leads to heuristics that completely fail at other scales (e.g., 5,000-item with $W$=500), so we used a varying-scale setup (Gao et al., 2024) that included data with different characteristics in the evaluation dataset $D$. The detailed settings of the evaluation datasets are exhibited in Table 6

**Pre-Trained Large Language Models**. The pre-trained LLM is *gpt-4o-mini-2024-07-18* for *GPT-4o-mini* and *gpt-3.5-turbo-0125* for *GPT-3.5-turbo*, we also use *Claude-3.5-sonnet-20241022* for *Claude-3.5-sonnet* and *DeepSeek-v3*, *Qwen2.5-Coder-32b-Instruct* in Appendix F.5.

*Table 6.* Compositions of the evaluation dataset $D$ on involved general frameworks and tasks.

| Framework | Step-by-step Construction | |
|---|---|---|
| Task | TSP | KP |
| Evaluation dataset $D$ | 64 50-node TSP instances | 64 100-item KP instances (W=25) |
| Framework | Step-by-step Construction | |
| Task | ASP | Online BPP |
| Evaluation dataset $D$ | 1 instance (n=15, w=10) | 4 WeiBull BPP instances
1,000-item instance with W=100
1,000-item instance with W=500
5,000-item instance with W=100
5,000-item instance with W=500 |
| Framework | ACO | |
| Task | TSP | CVRP |
| Evaluation dataset $D$ | 5 50-node TSP instances | 10 50-node CVRP instances |
| Framework | ACO | |
| Task | MKP | Offline BPP |
| Evaluation dataset $D$ | 5 100-item MKP instances (m=5) | 5 500-item BPP instances (W=150) |
| Framework | GLS | CAF Design |
| Task | TSP | - |
| Evaluation dataset $D$ | 10 TSP200 instances | 2 synthetic instances
The Ackley instance
The Rastrigin instance |

**Other Parameters.** MCTS-AHD adopts the same set of parameters for all tasks involved, and here we will summarize the other parameters of MCTS-AHD in the evaluation phase. The temperature of LLMs is fixed at 1.0. The number of initial nodes $N_I = 4$, the maximum depth of the MCTS tree $H = 10$, the number of mutations in each expansion $k = 2$, the maximum number of references in action e1 $p \in \{2, 3, 4, 5\}$, the initial exploration parameter $\lambda_0 = 0.1$, and the progressive widening parameter $\alpha = 0.5$. We consider $\lambda_0 = 0.1$ to be the most important setting of these and therefore ablate it in the main text, and we discuss the rest of the settings in Appendix F.9 to illustrate that none of these parameters are sensitive.

# E. Detailed Methodology

## E.1. Prompts of MCTS Actions

MCTS-AHD contains 6 distinct actions i1, e1, e2, m1, m2, and s1 for MCTS initialization and expansion. Next, we describe the meaning and prompt engineering of each action. These prompts contain the problem description, existing heuristic functions as contexts, function name, input name, and output name according to the task $P$ and the current MCTS. We execute the first LLM call through the following action prompts to obtain a design idea and its Python implementation for a heuristic function. The rest of this subsection provides examples for prompts, in which we highlight the unique part of each prompt in red colors (for all actions, we use the TSP task and the step-by-step construction framework as an example):

- **Initial Action i1**: Action i1 represents directly getting an idea of designing a valid heuristic function from scratch and a Python implementation directly through LLM.

> **Prompt for Action i1**
>
> Solving Traveling Salesman Problem (TSP) with constructive heuristics. TSP requires finding the shortest path that visits all given nodes and returns to the starting node.
>
> First, describe the design idea and main steps of your algorithm in one sentence. The description must be inside a brace outside the code implementation. Next, implement it in Python as a function named 'select_next_node'.
>
> This function should accept 4 input(s): 'current_node', 'destination_node', 'unvisited_nodes', 'distance_matrix'. The function should return 1 output(s): 'next_node'. The select_next_node function takes as input the current node, the destination_node, a set of unvisited nodes, and a distance matrix, and returns the next node to visit.
>
> Do not give additional explanations.

- **Crossover Action e1**: Action e1 inputs several distinct heuristic functions with their codes, their descriptions, and their performances. The prompt asks the LLM to get an idea for a new heuristic function different from all these heuristic functions and its corresponding Python implementation. The heuristics are selected from 2 to 5 subtrees of the MCTS root.

> **Prompt for Action e1**
>
> Solving Traveling Salesman Problem (TSP) with constructive heuristics. TSP requires finding the shortest path that visits all given nodes and returns to the starting node.
>
> I have k existing algorithms with their codes as follows:
> No.1 algorithm's description, its corresponding code and its objective value are:
> # Its Description
> # Its Python Code Implementation of A Function
> Objective value: # The Objective Value of the heuristic algorithm with the python code on a evaluation dataset.
> ...
> No.k algorithm's description, its corresponding code and its objective value are:
> # Its Description
> # Its Python Code Implementation of A Function
> Objective value: # The Objective Value of the heuristic algorithm with the python code on a evaluation dataset.
>
> Please create a new algorithm that has a totally different form from the given algorithms. Try generating codes with different structures, flows or algorithms. The new algorithm should have a relatively low objective value.
>
> First, describe the design idea and main steps of your algorithm in one sentence. The description must be inside

> a brace outside the code implementation. Next, implement it in Python as a function named 'select_next_node'.
>
> This function should accept 4 input(s): 'current_node', 'destination_node', 'unvisited_nodes', 'distance_matrix'. The function should return 1 output(s): 'next_node'. The select_next_node function takes as input the current node, the destination_node, a set of unvisited nodes, and a distance matrix, and returns the next node to visit.
>
> Do not give additional explanations.

- **Crossover Action e2**: Action e2 inputs one parent heuristic function (with its code, description, and performance) and one reference heuristic function sampled from a 10-size top-performing elite heuristic function set $E$ (we follow Liu et al. (2024b) in selecting a heuristic function from the set, the heuristic function in the set with rank $r_i$ will be sampled with a priority $p_i = \frac{1}{r_i + 10}$). The following prompt asks the LLM to design a new heuristic function based on the parent one and learn from the advantageous designs of the reference one.

> **Prompt for Action e2**
>
> Solving Traveling Salesman Problem (TSP) with constructive heuristics. TSP requires finding the shortest path that visits all given nodes and returns to the starting node.
>
> I have 2 existing algorithms with their codes as follows:
> No.1 algorithm's description, its corresponding code and its objective value are:
> # Its Description
> # Its Python Code Implementation of A Function
> Objective value: # The Objective Value of the heuristic algorithm with the python code on a evaluation dataset.
>
> No.2 algorithm's description, its corresponding code and its objective value are:
> # Its Description
> # Its Python Code Implementation of A Function
> Objective value: # The Objective Value of the heuristic algorithm with the python code on a evaluation dataset.
>
> Please create a new algorithm that has a similar form to the No.2 algorithm and is inspired by the No.1 algorithm. The new algorithm should have an objective value lower than both algorithms.
>
> Firstly, list the common ideas in the No.1 algorithm that may give good performances. Secondly, based on the common idea, describe the design idea based on the No.len(indivs) algorithm and main steps of your algorithm in one sentence. The description must be inside a brace. Next, implement it in Python as a function named 'select_next_node'.
>
> This function should accept 4 input(s): 'current_node', 'destination_node', 'unvisited_nodes', 'distance_matrix'. The function should return 1 output(s): 'next_node'. The select_next_node function takes as input the current node, the destination_node, a set of unvisited nodes, and a distance matrix, and returns the next node to visit.
>
> Do not give additional explanations.

- **Mutation Action m1**: Action m1 inputs a heuristic function with its description and code, it attempts to introduce more new mechanisms and new formulas or program segments to the input code through LLM.

> **Prompt for Action m1**
>
> Solving Traveling Salesman Problem (TSP) with constructive heuristics. TSP requires finding the shortest path that visits all given nodes and returns to the starting node.

I have one algorithm with its code as follows.:
# Its Description
# Its Python Code Implementation of A Function

Please create a new algorithm that has a different form but can be a modified version of the provided algorithm. Attempt to introduce more novel mechanisms and new equations or programme segments.

First, describe the design idea and main steps of your algorithm in one sentence. The description must be inside a brace outside the code implementation. Next, implement it in Python as a function named 'select_next_node'.

This function should accept 4 input(s): 'current_node', 'destination_node', 'unvisited_nodes', 'distance_matrix'. The function should return 1 output(s): 'next_node'. The select_next_node function takes as input the current node, the destination_node, a set of unvisited nodes, and a distance matrix, and returns the next node to visit.

Do not give additional explanations.

- **Mutation Action m2**: Action m2 also inputs the description and implementation of a heuristic function, it attempts to generate a heuristic function with different parameter settings through LLM.

**Prompt for Action m2**

Solving Traveling Salesman Problem (TSP) with constructive heuristics. TSP requires finding the shortest path that visits all given nodes and returns to the starting node.

I have one algorithm with its code as follows.:
# Its Description
# Its Python Code Implementation of A Function

Please identify the main algorithm parameters and help me in creating a new algorithm that has different parameter settings to equations compared to the provided algorithm.

First, describe the design idea and main steps of your algorithm in one sentence. The description must be inside a brace outside the code implementation. Next, implement it in Python as a function named 'select_next_node'.

This function should accept 4 input(s): 'current_node', 'destination_node', 'unvisited_nodes', 'distance_matrix'. The function should return 1 output(s): 'next_node'. The select_next_node function takes as input the current node, the destination_node, a set of unvisited nodes, and a distance matrix, and returns the next node to visit.

Do not give additional explanations.

- **Tree-Path Reasoning Action s1**: Action s1 is a tree-special action that takes all diverse heuristic functions (with their codes, descriptions, and performances) on a leaf-to-root tree path as input. The following prompt asks the LLM to get a better heuristic function based on these inputted in-contexts. When there is only one unique heuristic function in the tree path, we do not perform the action s1.

**Prompt for Action s1**

Solving Traveling Salesman Problem (TSP) with constructive heuristics. TSP requires finding the shortest path that visits all given nodes and returns to the starting node.

I have k existing algorithms with their codes as follows:

No.1 algorithm's description, its corresponding code and its objective value are:

# Its Description

# Its Python Code Implementation of A Function

Objective value: # The Objective Value of the heuristic algorithm with the python code on a evaluation dataset.

...

No.k algorithm's description, its corresponding code and its objective value are:

# Its Description

# Its Python Code Implementation of A Function

Objective value: # The Objective Value of the heuristic algorithm with the python code on a evaluation dataset.

Please help me create a new algorithm that is inspired by all the above algorithms with its objective value lower than any of them.

Firstly, list some ideas in the provided algorithms that are clearly helpful to a better algorithm. Secondly, based on the listed ideas, describe the design idea and main steps of your new algorithm in one sentence. The description must be inside a brace. Thirdly, implement it in Python as a function named 'select_next_node'.

This function should accept 4 input(s): 'current_node', 'destination_node', 'unvisited_nodes', 'distance_matrix'. The function should return 1 output(s): 'next_node'. The select_next_node function takes as input the current node, the destination_node, a set of unvisited nodes, and a distance matrix, and returns the next node to visit.

Do not give additional explanations.

## E.2. The Prompt of the Thought-Alignment Process

After obtaining the design idea and implementation by calling LLMs with the previous action prompts, MCTS-AHD will conduct the second LLM call. In the second LLM call, we will input the design idea and implementation obtained from the previous LLM call into the thought-alignment prompt as shown below for a more detailed and correlated heuristic function description.

**The thought-alignment Prompt**

Solving Traveling Salesman Problem (TSP) with constructive heuristics. TSP requires finding the shortest path that visits all given nodes and returns to the starting node.

Following is the Design Idea of a heuristic algorithm for the problem and the code with function name 'select_next_node' for implementing the heuristic algorithm.

The select_next_node function takes as input the current node, the destination_node, a set of unvisited nodes, and a distance matrix, and returns the next node to visit.

Design Idea:

# The Design Idea

Code:

# The Python Code Implementation of A Function

The content of the Design Idea idea cannot fully represent what the algorithm has done informative. So, now you should re-describe the algorithm using less than 3 sentences. Hint: You should reference the given Design Idea and highlight the most critical design ideas of the code. You can analyse the code to describe which variables are given higher priorities and which variables are given lower priorities, the parameters and the structure of the code.

### E.3. Examples of LLM Outputs

This subsection provides two examples of LLM output in designing heuristics with MCTS-AHD. Every single LLM-based heuristic generation includes two LLM calls. The first call generates **the Python code of a heuristic function** with action prompts and the second call generates **linguistic descriptions** for heuristic functions with thought-alignment prompts.

---

**Example of LLM output #1: Designing TSP heuristics within the step-by-step solving framework.**

**First Call Input:**
The prompt for an MCTS-AHD action.

**First Call Output:**
{The new algorithm will utilize a hybrid approach that weighs the immediate distance (1.2) more heavily while still considering the connection strength (0.6), ensuring that we select paths that are both cost-effective in the short term and reasonable in the longer term by optimizing the total accumulated cost.}

```python
def select_next_node(current_node, destination_node, unvisited_nodes,
    distance_matrix):
    if not unvisited_nodes:
 return None

    next_node = None
    min_combined_cost = float('inf')

    for candidate in unvisited_nodes:
 immediate_distance = distance_matrix[current_node][candidate]
 connection_strength = sum(distance_matrix[candidate][n] for n in unvisited_nodes if
     n != candidate) / (max(len(unvisited_nodes) - 1, 1))

 total_cost = immediate_distance
 current_candidate = candidate
 remaining_nodes = unvisited_nodes - {candidate}

 while remaining_nodes:
   best_next_node = min(remaining_nodes, key=lambda node: distance_matrix[
       current_candidate][node])
   total_cost += distance_matrix[current_candidate][best_next_node]
   current_candidate = best_next_node
   remaining_nodes.remove(best_next_node)

 total_cost += distance_matrix[current_candidate][destination_node]

 combined_cost = (1.2 * immediate_distance) + (0.6 / (connection_strength + 1)) +
     total_cost

 if combined_cost < min_combined_cost:
   min_combined_cost = combined_cost
   next_node = candidate

    return next_node
```

**Second Call Input:**
The prompt of the thought-alignment process.

**Second Call Output:**
The algorithm utilizes a hybrid approach to solve the Traveling Salesman Problem by prioritizing immediate distance (with a weight of 1.2) over connection strength (with a weight of 0.6) to select the next node to visit from a set of unvisited nodes. It calculates a combined cost that considers the immediate distance to a candidate node, the average connection strength to remaining unvisited nodes, and the total cost of visiting all remaining nodes before returning to the destination. The algorithm systematically evaluates candidate nodes based on this combined cost, ensuring a balance between short-term efficiency and overall path optimization.

---

## Example of LLM output #2: Designing CAF for Cost-aware BO.

**First Call Input:**
The prompt for an MCTS-AHD action.

**First Call Output:**
{The new algorithm evaluates the potential improvement over the best observed value while incorporating a compound risk factor based on both uncertainty and evaluation costs, adjusts the utility through a combination of historical performance scaling, and normalizes the results across the test inputs to favor those with high expected returns and low associated costs.}

```python
import torch

def utility(train_x, train_y, best_x, best_y, test_x, mean_test_y, std_test_y,
    cost_test_y, budget_used, budget_total):
    # Calculate remaining budget
    remaining_budget = budget_total - budget_used

    # Adjust the mean based on uncertainty
    risk_factor = std_test_y / (std_test_y + 1e-10)
    adjusted_mean = mean_test_y * (1 - risk_factor)

    # Improvement over the best observed value
    improvement_score = adjusted_mean - best_y

    # Calculate cost-effectiveness with exponential decay based on evaluation costs
    cost_effectiveness = (improvement_score / (cost_test_y + 1e-10)) * torch.exp(-
        cost_test_y / (remaining_budget + 1e-10))

    # Historical risk/variance for budget scaling
    historical_variance = torch.var(train_y, unbiased=False)
    budget_scaling = remaining_budget / (remaining_budget + historical_variance + 1e
        -10)

    # Combine cost-effectiveness with budget scaling
    final_utility = cost_effectiveness * budget_scaling

    # Normalize the utility values
    utility_value = (final_utility - torch.min(final_utility)) / (torch.max(
        final_utility) - torch.min(final_utility) + 1e-10)

    return utility_value
```

**Second Call Input:**
The prompt of the thought-alignment process.

**Second Call Output:**
The algorithm evaluates potential improvements over the best observed value by leveraging a risk-adjusted mean of the predicted function, where uncertainty is factored in through the standard deviation. It incorporates cost-effectiveness by weighing the expected improvement against evaluation costs, applying an exponential decay influenced by the remaining budget. Finally, the utility values are scaled by historical variance to prioritize inputs that promise higher expected returns while accounting for their associated costs, ultimately normalizing the output to facilitate comparison across test inputs.

### E.4. Total Algorithm

Algorithm 1 provides a pseudo-code for the proposed MCTS-AHD method, which provides details of the flow and parameter settings. The procedure of MCTS-AHD mainly includes an initialization process (up to line 8) and a four-stage MCTS process (line 8 to line 39).

---

**Algorithm 1** MCTS-AHD: Monte Carlo Tree Search for Automatic Heuristic Design

1: **Input:** Evaluation dataset $D$, The number of initial nodes $N_I$, Maximal evaluation times $T$, Maximal tree depth $H = 10$, Action set $\{i1, e1, e2, m1, m2, s1\}$, The number of mutation in each expansion $k = 2$, The number of subtrees in action e1 $\{2, 3, 4, 5\}$, UCT initial exploration parameter $\lambda_0 = 0.1$, Progressive widening parameter $\alpha = 0.5$.

2: **Output:** Code $C^*$ for the best found heuristic functions $h^*$.

3: Initialize a virtual root node $n_r$.      *// The virtual root node does not contain any codes.*

4: Set $t \leftarrow 0$.      *// $t$ represents the current number of evaluations.*

5: Initialize a code and its description with Action i1, and get all other $N_I - 1$ initial nodes with Action e1.

6: $q_{max} \leftarrow -1e5$, $q_{min} \leftarrow 0$

7: Link all the $N_I$ nodes to the MCTS root.

8: Evaluating the newly generated codes with evaluation dataset $D$, setting the Q, N values for these nodes.

9: **while** $t \leq T$ **do**

10:      $\lambda \leftarrow \lambda_0 * \frac{T-t}{T}$      *// **Exploration-decay.***

     *// MCTS Selection Stage: Selecting a node to expand*

11:      **if** $\lfloor N(n_r)^\alpha \rfloor \geq |\text{Children}(n_r)|$ **then**

12:          Expand $n_r$ with action e1, doing simulation and backpropagation.      *// **Progressive Widening for the root node.***

13:          $t \leftarrow t + 1$.      *// Updating evaluation time $t$.*

14:      **end if**

15:      $n_c \leftarrow n_r$      *// Selecting from root node.*

16:      **while** $n_c$ is not a leaf and its depth is less than $H$ **do**

17:          $n_c \leftarrow \arg\max_{c \in \text{Children}(n_c)} \left( \frac{Q(c) - q_{min}}{q_{max} - q_{min}} + \lambda \cdot \sqrt{\frac{\ln N(n_c + 1)}{N(c)}} \right)$

18:          **if** $\lfloor N(n_c)^\alpha \rfloor \geq |\text{Children}(n_c)|$ **then**

19:              Expand $n_c$ with action e2, doing simulation and backpropagation.      *// **Progressive Widening non-root nodes.***

20:              $t \leftarrow t + 1$.      *// Updating evaluation time $t$.*

21:          **end if**

22:      **end while**

     *// MCTS Expansion Stage: Add new nodes to the tree*

23:      Conducting action e2, s1, $k$ times of m1, and $k$ times of m2 to the $n_c$ by LLM.

     *// MCTS Simulation Stage: Get Q values for newly expanded nodes*

24:      Evaluating the newly generated heuristic function with evaluation dataset $D$,

25:      Setting the Q, N values for these nodes.

26:      If a better heuristic function is found, update the best found code $C^*$ with its implementation.

27:      Updating the elite set $E$ if new top 10 heuristic functions emerge.

28:      $t \leftarrow t + 2k + 2$.      *// Updating the number of evaluations $t$.*

29:      **for** $c \in \text{Children}(n_c)$ **do**

30:          $q_{max} \leftarrow \max(q_{max}, Q(c))$, $q_{min} \leftarrow \min(q_{min}, Q(c))$      *// Updating $q_{max}$ and $q_{min}$.*

31:      **end for**

     *// MCTS Backpropagation Stage: Update Q and N values*

32:      **while** $n_c$ is not the root $n_r$ **do**

33:          $Q(n_c) \leftarrow \max_{c \in \text{Children}(n_c)} Q(c)$      *// Updating the Q value.*

34:          $N(n_c) \leftarrow N(n_c) + 2k + 2$      *// Updating visit times.*

35:          $n_c \leftarrow \text{Father}(n_c)$

36:      **end while**

37:      $Q(n_r) \leftarrow \max_{c \in \text{Children}(n_r)} Q(c)$      *// Updating the Q value.*

38:      $N(n_r) \leftarrow N(n_r) + 2k + 2$      *// Updating visit times.*

39: **end while**

40: **Return:** The best found code $C^*$ for heuristics.

---

## F. Experiment Details

### F.1. Designing Heuristics with the Step-by-step Construction Framework for ASP

We apply MCTS-AHD to another NP-hard CO problem ASP (Romera-Paredes et al., 2024). ASP constructs a set of n-dimensional vectors $\mathcal{A}$ (named an admissible set) in which each vector $\in \{0, 1, 2\}^n$ and the number of non-zero items in each vector is constrained to be $w$. ASP aims to maximize the size of the admissible set $\mathcal{A}$ with another certain constraint between the vectors in the admissible set (detailed in Appendix B.1). LLM-based AHD methods are responsible for designing a priority function: $\{0, 1, 2\}^n \to \mathbb{R}$, which is executed iteratively to construct $\mathcal{A}$ step by step. We use a single instance with $n$=15 and $w$=10 as the evaluation dataset $D$. The experiments in Table 7 test the heuristics on four different scales where MCTS-AHD also exhibits the best results compared to other existing LLM-based AHD methods in most test sets.

*Table 7.* Designing step-by-step construction heuristics for ASP. Each LLM-based AHD method is run three times for its average performance. The size of $\mathcal{A}$ has a theoretically upper bound $\binom{n}{w}$, and we take this value as Optimal. The test set of each scale contains only one instance and we underline the in-domain scale. The best-performing method for each LLM model is shaded and the overall best result is in bold.

| Task | ASP | | | |
|---|---|---|---|---|
| Methods | $n$=12,$w$=7 | $n$=15,$w$=10: | $n$=21,$w$=15: | $n$=24,$w$=17: |
| Optimal | 792 | 3003 | 43596 | 237984 |
| LLM-based AHD: *GPT-3.5-turbo* | | | | |
| Funsearch | 612.0 | 2057.0 | 10664.0 | 37323.0 |
| EoH | 772.0 | 2759.0 | 30869.3 | 147323.0 |
| MCTS-AHD (Ours) | 784.0 | 2784.0 | 30608.0 | 150729.0 |
| LLM-based AHD: *GPT-4o-mini* | | | | |
| Funsearch | 622.0 | 2410.0 | 10758.7 | 43217.0 |
| ReEvo | 784.0 | 2733.0 | 25753.7 | 115709.0 |
| HSEvo | 744.0 | 2307.0 | 18756.3 | 81185.0 |
| EoH | 779.0 | 2776.0 | 32716.7 | 160024.0 |
| MCTS-AHD (Ours) | 775.0 | 2780.0 | 32900.3 | 163832.0 |

### F.2. Guided Local Search Framework

To evaluate the performance of MCTS-AHD in designing the penalty heuristics of the GLS framework, we follow Ye et al. (2024a) in using MCTS-AHD for the key heuristic function to generate the knowledge matrix of KGLS (Arnold et al., 2019) and employing 10 TSP200 instances for performance evaluations (as $D$). The number of local search iterations in GLS is set to 1200 for both heuristic designs and heuristic testings. We compare MCTS-AHD to LLM-based AHD baselines, NCO methods with GLS frameworks, and the original KGLS with manually designed heuristic functions. As shown in Table 8, the proposed MCTS-AHD can refine the KGLS on both the TSP200 and TSP500 test sets and MCTS-AHD generally outperforms other LLM-based AHD methods using *GPT-4o-mini* as the black-box pre-trained LLM.

*Table 8.* Designing heuristics within the GLS general framework for solving TSP. KGLS, NCO methods, and heuristics from LLM-based AHD allow 1200 iterations of local search. The test sets of TSP with $N$=100 and $N$=200 are 1,000-instance test sets from Table 1 and for TSP500 and TSP1000, we prepare two 64-instance test sets. For NeuralGLS* and GNNGLS*, we use the results reported in Ye et al. (2024a). The table shows the gaps to optimal and each LLM-based AHD method is run three times for the average optimality gaps.

| | TSP-GLS | | | |
|---|---|---|---|---|
| N= | 100 | 200 | 500 | 1,000 |
| Optimal | 0.0000% | 0.0000% | 0.0000% | 0.0000% |
| KGLS | 0.0034% | 0.2270% | 0.9578% | 1.5348% |
| *NCO methods with the GLS general framework* | | | | |
| VRP-DACT (Ma et al., 2021) | 1.7943% | 91.9267% | - | - |
| NeuOpt (Ma et al., 2023) | 0.2950% | 0.9152% | - | - |
| NeuralGLS* (Sui et al., 2024) | 0.470%* | 3.622%* | - | - |
| GNNGLS* (Hudson et al., 2021) | 0.705%* | 3.522%* | - | - |
| *LLM-based AHD: GPT-4o-mini* | | | | |
| EoH | 0.0065% | 0.2025% | 0.9534% | 1.6083% |
| ReEvo | 0.0076% | 0.2210% | 0.9993% | 1.6155% |
| MCTS-AHD (Ours) | 0.0060% | 0.2106% | 0.9495% | 1.5985% |

## F.3. Time Consumption of MCTS-AHD Heuristic Evolution

Compared to EoH and ReEvo, MCTS-AHD will not cause server efficiency degradation regarding time and token consumption. To demonstrate this, as shown in Table 9, we provide a detailed comparison of the time and token consumption of MCTS-AHD and population-based baselines in five application scenarios. We calculate the token number based on *GPT-4o-mini*. Results demonstrate that compared to population-based methods, MCTS-AHD has a slight efficiency decrease and maintains a similar level of token consumption compared to LLM-based AHD baselines.

*Table 9.* Time and token consumption in designing heuristics with different algorithms.

| Methods | Frameworks | Construction | | ACO | | BO |
|---|---|---|---|---|---|---|
| | Problems | TSP | KP | TSP | MKP | CAF |
| EoH | Time | 2h | 2h | 4h | 4h | 15h |
| | Input Token | 0.8M | 0.7M | 1M | 1M | 1.2M |
| | Output Token | 0.2M | 0.2M | 0.5M | 0.5M | 0.5M |
| ReEvo | Time | 2h | - | 5h | 5h | - |
| | Input Token | 1.1M | - | 1.3M | 1.3M | - |
| | Output Token | 0.4M | - | 0.6M | 0.5M | - |
| MCTS-AHD | Time | 4h | 3h | 8h | 4h | 14h |
| | Input Token | 1M | 1M | 1.2M | 1.3M | 1.3M |
| | Output Token | 0.3M | 0.2M | 0.5M | 0.6M | 0.6M |

## F.4. P-values for Significance

The design performance of the LLM-based AHD method follows an implicit distribution. Although running a method three times can preliminarily indicate the advantages and disadvantages of this method, it still cannot prove the significant difference between its performance distribution algorithm and the performance distributions of comparison algorithms. In this section, we introduce the p-value to demonstrate the significant advantage of MCT-AHD compared to the main LLM-based AHD baseline EoH. We employ both EoH and MCTS-AHD to design up to ten heuristics for a portion of CO problems considered in this paper, and the results and p-values for each run are shown in Table 10. In any of the four application scenarios, there is at least a 96% confidence in satisfying the hypothesis that MCTS-AHD leads compared to an LLM-based AHD baseline with population EoH.

*Table 10.* Up to ten runs of EoH and MCTS-AHD on a portion of NP-hard CO problems with step-by-step construction frameworks and ACO frameworks. "avg" in the Table below represents the average and "std" means the standard variant. The p-value is calculated with single-tailed t-tests.

| CO Problem | Methods | run1 | run2 | run3 | run4 | run5 | run6 | run7 | run8 | run9 | run10 | avg | std | p-value |
|---|---|---|---|---|---|---|---|---|---|---|---|---|---|---|
| | | | | | General Framework: Step-by-step Construction, LLM: *GPT-4o-mini* | | | | | | | | | |
| TSP50 | EoH | 6.452 | 6.447 | 6.284 | 6.386 | 6.316 | 6.372 | 6.480 | 6.480 | 6.259 | 6.388 | 6.386 | 0.080 | 0.002855655 |
| | ReEvo | 6.3592 | 6.2681 | 6.4516 | 6.3841 | 6.711 | 6.437 | 6.272 | 6.297 | 6.388 | 6.431 | 6.400 | 0.128 | 0.009529994 |
| | HSEvo | 6.4582 | 6.4468 | 6.2839 | 6.360 | 6.461 | 6.446 | 6.400 | 6.285 | 6.452 | 6.376 | 6.397 | 0.069 | 0.000796727 |
| | MCTS-AHD | 6.174 | 6.156 | 6.347 | 6.356 | 6.285 | 6.274 | 6.257 | 6.365 | 6.289 | 6.302 | 6.280 | 0.071 | |
| KP100, W=25 | EoH | 40.229 | 40.231 | 40.232 | 40.231 | 40.234 | 40.264 | 40.240 | 40.235 | 40.229 | 40.235 | 40.236 | 0.010 | 0.027524885 |
| | MCTS-AHD | 40.259 | 40.265 | 40.231 | 40.236 | 40.233 | 40.262 | 40.264 | 40.233 | 40.233 | 40.262 | 40.248 | 0.015 | |
| | | | | | General Framework: ACO, LLM: *GPT-4o-mini* | | | | | | | | | |
| TSP50 | EoH | 5.827 | 5.825 | 5.831 | 5.830 | 5.828 | - | - | - | - | - | 5.828 | 0.003 | 0.039230447 |
| | ReEvo | 5.894 | 5.829 | 5.844 | 5.831 | 5.820 | - | - | - | - | - | 5.844 | 0.029 | 0.02477005 |
| | HSEvo | 5.827 | 5.802 | 5.799 | 5.823 | 5.843 | - | - | - | - | - | 5.819 | 0.018 | 0.115934978 |
| | MCTS-AHD | 5.798 | 5.779 | 5.827 | 5.742 | 5.830 | - | - | - | - | - | 5.795 | 0.036 | |
| MKP100, m=5 | EoH | 23.149 | 23.133 | 23.136 | 23.311 | 23.266 | - | - | - | - | - | 23.199 | 0.083 | 0.037268388 |
| | ReEvo | 23.310 | 23.187 | 23.237 | 23.291 | 23.263 | - | - | - | - | - | 23.258 | 0.048 | 0.20630356 |
| | HSEvo | 23.280 | 23.270 | 23.279 | 23.268 | 23.286 | - | - | - | - | - | 23.277 | 0.008 | 0.433650083 |
| | MCTS-AHD | 23.235 | 23.284 | 23.287 | 23.294 | 23.294 | - | - | - | - | - | 23.279 | 0.025 | |

## F.5. MCTS-AHD with Other LLMs

Besides using GPT models as LLMs (as shown in the other part of this article), MCTS-AHD can cooperate with other open-source or closed-source LLMs as well. This subsection evaluates MCTS-AHD on designing step-by-step construction heuristics for TSP and KP utilizing other advanced pre-trained LLMs, including closed-source LLM *Claude-3.5-sonnet* and Open-source LLM *DeepSeek-v3*, *Qwen2.5-Coder-32b-Instruct*. Performances of the designed heuristics are shown in Table 11, where heuristics design with *Claude-3.5-sonnet*, *DeepSeek-v3*, and *Qwen2.5-Coder-32b-Instruct* can still surpass handcrafted heuristics in both TSP and KP test sets and outperform the advanced NCO method POMO (which needs task-specific training) in KP. Moreover, MCTS-AHD runs with *GPT-4o-mini* exhibit the best performance in each test set, demonstrating that *GPT-4o mini* could be the best choice in LLM for MCTS-AHD.

*Table 11.* An extension of Table 1, employing both GPT models and open-source LLMs for MCTS-AHD. Each MCTS-AHD result is averaged over three runs and we highlight the MCTS-AHD result with the best performance on each test set.

| | TSP | | | | KP | | | |
|---|---|---|---|---|---|---|---|---|
| N= | *N*=50 | | *N*=100 | | *N*=100, *W*=25 | | *N*=200, *W*=25 | |
| Method | Obj.↓ | Gap | Obj.↓ | Gap | Obj.↑ | Gap | Obj.↑ | Gap |
| Optimal | 5.675 | - | 7.768 | - | 40.271 | - | 57.448 | - |
| Greedy Construct | 6.959 | 22.62% | 9.706 | 24.94% | 40.225 | 0.12% | 57.395 | 0.09% |
| POMO | 5.697 | 0.39% | 8.001 | 3.01% | 39.676 | 1.48% | 57.271 | 0.09% |
| Closed-Source LLMs | | | | | | | | |
| MCTS-AHD(*GPT-3.5-turbo*) | 6.346 | 11.82% | 8.861 | 14.08% | 40.233 | 0.09% | 57.393 | 0.10% |
| MCTS-AHD(*GPT-4o-mini*) | 6.225 | 9.69% | 8.684 | 11.79% | 40.252 | 0.05% | 57.423 | 0.04% |
| MCTS-AHD(*Claude-3.5-sonnet*) | 6.503 | 14.57% | 9.036 | 16.32% | 40.233 | 0.10% | 57.396 | 0.09% |
| Open-Source LLMs | | | | | | | | |
| MCTS-AHD(*DeepSeek-v3*) | 6.348 | 11.85% | 8.859 | 14.04% | 40.233 | 0.10% | 57.402 | 0.08% |
| MCTS-AHD(*Qwen2.5-Coder-32b-Instruct*) | 6.272 | 10.52% | 8.685 | 11.81% | 40.235 | 0.09% | 57.402 | 0.08% |

## F.6. Application on General Optimization Tasks

In this section, we discuss the application of MCTS-AHD on more optimization problems. As an AHD method, MCTS-AHD aims at designing and evolving high-quality heuristics for optimization problems within a given solving framework. There are no heuristics for some optimization problems (e.g., code search (DeLorenzo et al., 2024)), so MCTS-AHD cannot be directly applied to them. When directly implementing MCTS-AHD to code search (e.g., the APPS dataset (Hendrycks et al., 2021)), the current MCTS-AHD cannot handle code bugs and becomes ineffective.

To further assess the generalizability of MCTS-AHD across different domains. We evaluate MCTS-AHD on a policy optimization task *MountainCar-v0* based on the Gym framework. Table 12 shows the average steps needed to reach the goal, where the heuristic policy designed by MCTS-AHD is more effective. We run AHD methods three times for the average performance.

*Table 12.* Results of designing heuristics for MountainCar-v0.

| | EoH | ReEvo | MCTS-AHD |
|---|---|---|---|
| Gym-MountainCar-v0 | 140.3 | 117.6 | **115.0** |

In conclusion, MCTS-AHD demonstrates superiority in designing heuristics for various optimization problems, including combinatorial optimization problems, Bayesian optimization tasks, and policy optimization tasks using the same set of hyperparameters. We believe MCTS-AHD can be useful in a wide range of application scenarios.

## F.7. Results on TSPLib: Compare to GP-based AHD Methods

We conduct tests on a well-known real-world TSP benchmark TSPLib (Reinelt, 1991) (we adopt instances with nodes less than 500 in this article) to compare the quality of MCTS-AHD to a GP-based AHD method GHPP (Duflo et al., 2019). Christofides (Christofides, 2022), Greedy (Brecklinghaus & Hougardy, 2015), Nearest insertion, and nearest-greedy (Rosenkrantz et al., 1977) are famous manually designed heuristics for TSP. We use the reported results of these algorithms in the article (Duflo et al., 2019). Meanwhile, the results of GHPP also come from Duflo et al. (2019).

For LLM-based AHD methods EoH and MCTS-AHD, we use the best-performing step-by-step constructive heuristic among their three runs with *GPT-4o-mini* for their performances. We use the best-performing heuristic of ReEvo according to their report in Ye et al. (2024a). On each TSPLib instance, We run the heuristics of EoH, ReEvo, and MCTS-AHD three times with different starting nodes for an average performance. As shown in Table 13, the heuristic from MCTS-AHD can surpass manually designed baselines, the GP-based AHD method GHPP, and LLM-based AHD baselines in the average optimality gap.

*Table 13.* Results of GP-based AHD method GPHH, LLM-based methods on designing heuristics for TSP with the step-by-step construction framework. Christofides, Greedy, Nearest insertion, and Nearest-greedy are manually designed heuristics for TSP where their results are also drawn from (Duflo et al., 2019). We report the optimality gap of each instance and heuristics designed by LLM-based AHD methods are run 3 times with different starting nodes for average performances. The leading LLM-designed heuristic on each instance is marked in shaded and the overall best heuristic is in bold.

| Instance | Christofides | Greedy | Nearest insertion | Nearest-greedy | GPHH-best | EoH | ReEvo | MCTS-AHD |
|---|---|---|---|---|---|---|---|---|
| ts225.tsp | 5.67% | **5.38%** | 19.93% | 16.82% | 7.71% | 5.57% | 6.56% | 10.84% |
| rat99.tsp | **9.43%** | 22.30% | 21.05% | 21.79% | 14.09% | 18.78% | 12.41% | 10.46% |
| bier127.tsp | 13.03% | 19.50% | 23.05% | 23.25% | 15.64% | 14.05% | 10.79% | **7.56%** |
| lin318.tsp | 13.80% | 18.75% | 24.44% | 25.78% | 14.30% | **14.03%** | 16.63% | 14.07% |
| eil51.tsp | 15.18% | 13.03% | 16.14% | 31.96% | 10.20% | 8.37% | **6.47%** | 15.98% |
| d493.tsp | **9.52%** | 16.68% | 20.39% | 24.00% | 15.58% | 12.41% | 13.43% | 11.73% |
| kroB100.tsp | **9.82%** | 16.59% | 21.53% | 26.26% | 14.06% | 13.46% | 12.20% | 11.43% |
| kroC100.tsp | 9.08% | 12.94% | 24.25% | 25.76% | 16.22% | 16.85% | 15.88% | **8.27%** |
| ch130.tsp | 10.09% | 28.40% | 19.21% | 25.66% | 14.77% | 12.26% | **9.40%** | 10.18% |
| pr299.tsp | **11.23%** | 31.42% | 25.05% | 31.42% | 18.24% | 23.58% | 20.63% | **11.23%** |
| fl417.tsp | 15.57% | 12.64% | 25.52% | 32.42% | 22.72% | 20.47% | 19.15% | **10.20%** |
| kroA150.tsp | 13.44% | 20.24% | 19.09% | 26.08% | 15.59% | 18.36% | 11.62% | **10.08%** |
| pr264.tsp | **11.28%** | 11.89% | 34.28% | 17.87% | 23.96% | 18.03% | 16.78% | 12.27% |
| pr226.tsp | 14.17% | 21.44% | 28.02% | 24.65% | 15.51% | 19.90% | 18.02% | **7.15%** |
| pr439.tsp | **11.16%** | 20.08% | 24.67% | 27.36% | 21.36% | 21.96% | 19.25% | 15.12% |
| Average Gap | 11.50% | 18.09% | 23.11% | 25.41% | 16.00% | 15.87% | 13.95% | **11.10%** |

## F.8. Compare to LLM-as-Optimizer Methods

As discussed in Appendix A, another LLM-based approach for CO problems, LLM-as-optimizer methods cannot achieve outstanding performance in large-scale instances. Here we compare this type of method with the proposed MCTS-AHD. As shown in Table 14, LLM-as-optimizer methods LEMA (Liu et al., 2024e) and OPRO (Yang et al., 2024) can provide better solutions in very-small-scale TSP20 instances, but as the scale increases to TSP50, these methods will fail to achieve convergence in LLM-based solution optimizations.

*Table 14.* Comparison of LLM-as-optimizer methods and MCTS-AHD designed heuristics. We display the optimality gap of MCTS-AHD on TSP20 and TSP50 test sets with 1,000 instances, respectively. Results of LEMA and OPRO are drawn from the original literature and the LLM for all methods in this table is *GPT-3.5-turbo*.

| Methods | LEMA* | OPRO* | MCTS-AHD(step-by-step construction) |
|---|---|---|---|
| TSP20 | 3.94% | 4.40% | 7.71% |
| TSP50 | - | 133.00% | 11.82% |

## F.9. Discussion: Ablation of Other Parameters

In 5.1, we have provided partial ablation experiments, focusing mainly on verifying that $\lambda - 0 = 0.1$ is a reasonable setting and validating the effectiveness of the three proposed components (Progressive Widening, Thought Alignment, and Exploration decay) and MCTS-AHD's expansion actions. MCTS-AHD also includes other parameters, such as the number of initial solutions $N_I$, the threshold parameter for progressive widening $\alpha$, and the number of actions m1 and m2 in each expansion $k$. We believe that MCTS-AHD is not sensitive to the settings of these parameters. In this section, we will introduce the reasons for their default settings and demonstrate their flexibility through ablation experiments.

$N_I$ determines the number of initial nodes (i.e., heuristic samples), which affects the perception of heuristic space by LLM in the early iterations of the heuristic evolution. Table 15 shows that adjusting it to $N_I = 10$ has no significant effect on the results, indicating that using $N_I = 4$ is sufficient. Under the setting of evaluating $T$ times in total, progressive widening allows the maximum number of level-1 tree nodes connected to the root to be at most $\lfloor T^\alpha \rfloor$. Larger $\alpha$ settings will drive the MCTS tree shallow, affecting the quality of multi-hop reasoning brought by MCTS. Therefore, we choose to set $\alpha = 0.5$ to balance the depth of the MCTS tree and the importance of progressive widening. According to Table 15, changing this setting to $\alpha = 0.6$ does not result in significant effect degradation. The setting of $k = 2$ allows MCTS to utilize the randomness brought about by LLM in trying different search directions in each expansion, and setting it to $k = 1$ does not cause a significant degradation in performances as well.

*Table 15.* Ablation on the number of initial solution $N_I$, the threshold parameter for progressive widening $\alpha$, the number of actions m1 and m2 in each expansion $k$. The performances of MCTS-AHD variants are averaged over five runs.

|  | TSP50 | KP100 |
| --- | --- | --- |
| $N_I = 4$ (Default Setting in MCTS-AHD, 10 runs) | 10.661% | 0.059% |
| $N_I = 10$ | 11.094% | 0.047% |
| $\alpha = 0.5$ (Default Setting in MCTS-AHD, 10 runs) | 10.661% | 0.059% |
| $\alpha = 0.6$ | 10.891% | 0.063% |
| $k = 2$ (Default Setting in MCTS-AHD, 10 runs) | 10.661% | 0.059% |
| $k = 1$ | 11.380% | 0.049% |

## F.10. Discussion: The Advantage Scope of MCTS

MCTS-AHD shows greater strengths in application scenarios with a more complex heuristic space $H$ and tasks with more descriptions as knowledge.

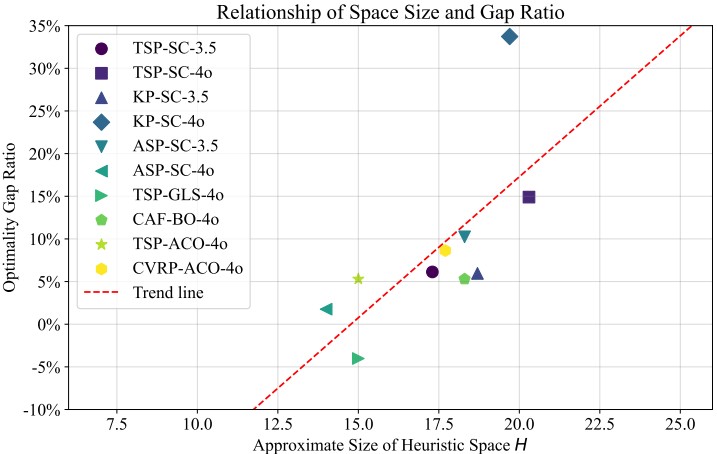

*Figure 5.* The relation of approximate heuristic space size $n$ and the optimality gap ratio between EoH and MCTS-AHD (i.e., $1 - \text{Gap}_{MCTS-AHD}/\text{Gap}_{EoH}$). We consider EoH as a baseline for its applicability in all tasks. We do not include tasks with unavailable optimal objective values (e.g., online BPP). The legends have been simplified, for example, TSP-GLS-4o represents designing GLS heuristics for TSP with *GPT-4o-mini*. SC in the figure is an abbreviation for step-by-step construction.

- **Better in Application Scenarios with a More Complex Heuristic Space $H$.** Each heuristic function can be expressed in the form of $a_1 f_1(x) + a_2 f_2(x) + a_3 f_3(x) + \ldots + a_n f_n(x)$, representing linear combinations of different sub-

functions, where $x$ denotes the particular input ($\boldsymbol{ins}$) of the heuristic function $h$. The heuristic space $H$ to be explored for an application scenario can be defined as the set of all meaningful sub-functions, i.e., $H = \text{Span}\{f_1(x), \ldots\}$. To estimate the size of the set consisting of all meaningful sub-functions, we use OpenAI *o1-mini-2024-09-12* to analyze the top 10 heuristic functions of EoH and MCTS-AHD in their respective 3 runs (6 runs and 60 heuristic functions in total), ordering LLM to break functions down into linearly independent sub-functions and use the number of sub-functions $n$ to estimate the size of the heuristic space.

We hypothesize that for more complex heuristic spaces, MCTS-AHD can explore the heuristic space more comprehensively compared to population-based methods such as EoH. To test this hypothesis, in Figure 5, we plot the relation of estimated heuristic space size $n$ and the leads in optimality gap between MCTS-AHD and EoH (the y-axis in Figure, $1 - \text{Gap}_{MCTS-AHD}/\text{Gap}_{EoH}$). The results verify our hypothesis. As the trend line demonstrates, MCTS-AHD tends to achieve a more significant lead compared to the population-based baseline EoH in application scenarios with larger $n$. It indicates that MCTS-AHD may be more suitable for application scenarios with more complex heuristic spaces $H$.

- **Better in Application Scenarios with More Descriptions**. As shown in Table 16, MCTS-AHD demonstrates a significant decrease in effectiveness in black-box settings, taking the lead only in the TSP task. This suggests that MCTS-AHD will perform better in application scenarios with more descriptions (e.g.,white-box cases).

*Table 16.* Implementing MCTS-AHD on Black-box CO tasks with ACO general frameworks. We follow the settings of Ye et al. (2024a) in heuristic evolution and run each LLM-based method three times for average performance. The white-box results are the same as Table 2.

| | TSP | CVRP | MKP | Offline BPP |
|---|---|---|---|---|
| N= | $N$=50 | $N$=50, $C$=50 | $N$=100,$m$=5 | $N$=500,$C$=150 |
| Methods | Obj.↓ | Obj.↓ | Obj.↑ | Obj.↓ |
| ACO | 5.992 | 11.355 | 22.738 | 208.828 |
| DeepACO | 5.842 | 8.888 | 23.093 | 203.125 |
| White-box Setting: *GPT-4o-mini* | | | | |
| EoH | 5.828 | 9.359 | 23.139 | 204.646 |
| ReEvo | 5.856 | 9.327 | 23.245 | 206.693 |
| MCTS-AHD(Ours) | 5.801 | 9.286 | 23.269 | 204.094 |
| Black-box Setting: *GPT-4o-mini* | | | | |
| EoH | 5.831 | 9.401 | 23.240 | 204.615 |
| ReEvo | 5.860 | 9.404 | 23.196 | 206.021 |
| MCTS-AHD(Ours) | 5.830 | 9.444 | 23.191 | 205.375 |

We can explain that MCTS-AHD's better performance in application scenarios with complex heuristic space is highly beneficial from its ability to explore complex function spaces and escape from local optima.

For its mediocre performance in black-box application scenarios, I believe this is mainly because compared to MCTS, which only performs limited expansion on each heuristic, the population structure is computationally intensive and often involves many rounds of LLM-based operations on a heuristic function in the population. So, the effectiveness of MCTS-AHD may require LLMs more on its generation quality in limited number of MCTS expansions. In contrast, black-box application scenarios make it difficult for LLMs to guarantee this condition, so these application scenarios will be tough for MCTS-AHD.

## G. Examples of Evolution

To visually demonstrate the workflow of MCTS-AHD and its ability on exploration, we provide two examples of the evolution of heuristics in two tasks, as shown in Figure 6.

The two examples of evolution clearly exhibit how MCTS-AHD conducts comprehensive explorations. For example, in designing heuristics with a step-by-step construction framework for TSP, MCTS-AHD can expand potential child nodes from nodes (e.g., MCTS node with "Expansion: t=611") that are not among the top 10 optimal ones (the performance range of the top 10 optimal heuristics is the yellow shade), and ultimately reach the best heuristic. It reflects the superiority of employing MCTS as an optimization framework instead of population-based EC. Population-based LLM-based AHD methods such as EoH, ReEvo, and HSEvo lack consideration of worse-performing but potential heuristics. These algorithms will be obsessed with processing top-performance heuristic functions. When the top 10 heuristics cannot obtain new heuristics surpassing the top 10 performances with one LLM-based operation on heuristics, the evolution of heuristics will trap into local optima.

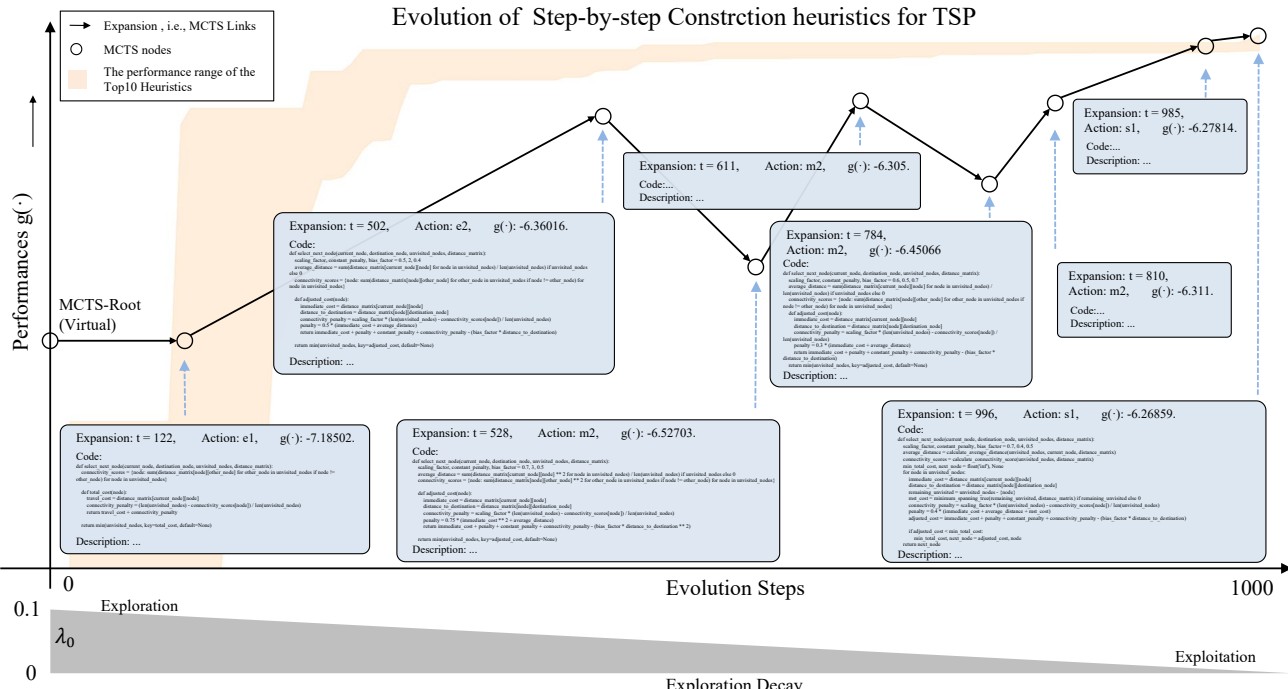

(a) An Example of MCTS-AHD Evolution on Designing Step-by-step Construction Heuristics for TSP

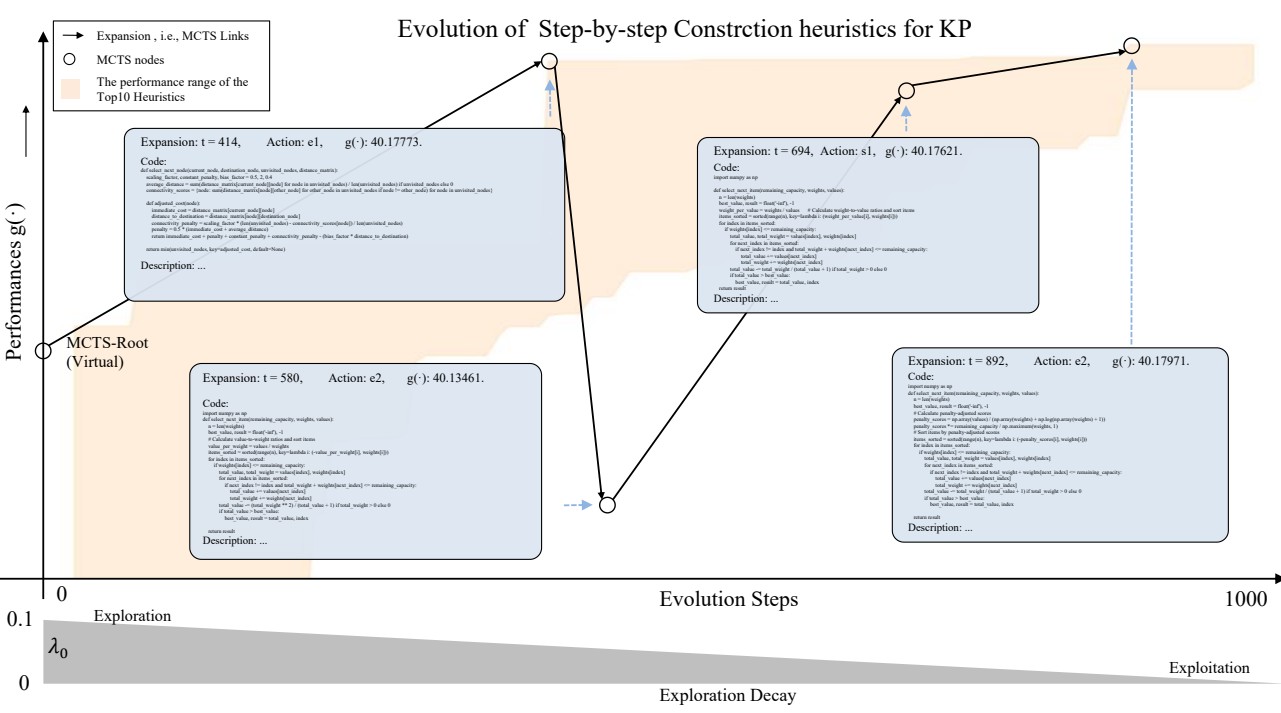

(b) An Example of MCTS-AHD Evolution on Designing Step-by-step Construction Heuristics for KP

*Figure 6.* Two examples of the heuristic evolution process of MCTS-AHD. We provide the performance function value $g(\cdot)$ of the heuristic function within each MCTS tree node. The "Action" in each node represents one of the six actions of expanding this node. Codes are simplified by OpenAI *GPT* to reduce the space occupations.

## H. Baselines & Licenses

**For the optimal value** in Table 1 and Table 8. We run LKH3 for TSP instances with adopting the commonly used setting in NCO methods (Kool et al., 2018). We set the LKH parameters RUNS = 10 and MAX_TRAILS = 1,0000. For KP instances, we use Google OR-Tools for the optimal value.

**For manually designed heuristics with general frameworks**, this paper includes Greedy Construct, the ACO algorithm (Dorigo et al., 2006), the KGLS algorithm (Arnold & Sörensen, 2019), and EI (Mockus, 1974), EIpu (Snoek et al., 2012), EI-cools (Lee et al., 2020a) for the CAF design task in BO. We implement KGSL by setting the heuristic matrix to the distance matrix, using the implementation of ACO in DeepACO (Ye et al., 2024b), and use the results reported in Yao et al. (2024c) for EI, EIpu, and EI-cools.

**For NCO baselines**, this article implements POMO (Kwon et al., 2020), DeepACO (Ye et al., 2024b), VRP-DACT (Ma et al., 2021), and NeuOpt (Ma et al., 2023). For a fair comparison, in Table 1, the reported POMO solutions are from a single start and generated without augmentations. The maximum operation on a solution is set to $T = 1200$ for VRP-DACT and NeuOpt.

**For AHD baselines**, in TSPLib, we adopt the results of GHPP reported in (Duflo et al., 2019).

**For LLM-based AHD baselines**, this article considers Funsearch (Romera-Paredes et al., 2024), EoH (Liu et al., 2024b), ReEvo (Ye et al., 2024a), and HSEvo (Dat et al., 2024), we follow all their parameter settings in evolutions (e.g., setting population size $M = 20$ for online BPP and $M = 10$ for other tasks). We try not to introduce too much external expert information. However, in implementations, Funsearch needs to maintain 10 relatively independent multiple populations from a certain inferior seed heuristic function. ReEvo relies more on the seed heuristic function, and the ReEvo method will become unavailable when all individuals in the elite population have the same objective function value. So, in some application scenarios (e.g., step-by-step construction for the KP task), it is too hard to find a good seed heuristic function that ensures the availability of the algorithm while trying not to provide too much external knowledge.

In experiments, we use the same seed functions for baselines (Funsearch, ReEvo, and HSEvo), using seed functions proposed in Ye et al. (2024a) for ACO frameworks and GLS frameworks, random selection functions for step-by-step constructing TSP and ASP, and the best-known function proposed in Romera-Paredes et al. (2024) for online BPP. The proposed MCTS-AHD, exhibits better applicability and superior performance in most application scenarios, without the requirement of seed function design. In our seed function settings, ReEvo and its follow-up method HSEvo are not available in some application scenarios (e.g., KP, ACO), where we do not report their effects.

### H.1. License

The licenses and URL of baselines are listed in Table 17.

*Table 17.* A summary of licenses.

| Resources | Type | License | URL |
|---|---|---|---|
| LKH3 | Code | Available for academic research use | http://webhotel4.ruc.dk/~keld/research/LKH-3/ |
| OR-Tools | Code | MIT License | https://developers.google.com/optimization/pack/knapsack?hl=zh-cn |
| POMO | Code | Available online | https://github.com/yd-kwon/POMO/tree/master |
| DeepACO | Code | MIT License | https://github.com/henry-yeh/DeepACO |
| VRP-DACT | Code | MIT License | https://github.com/yining043/VRP-DACT |
| NeuOpt | Code | MIT License | https://github.com/yining043/NeuOpt |
| Funsearch | Code | Apache License | https://github.com/google-deepmind/funsearch |
| EoH | Code | MIT License | https://github.com/FeiLiu36/EoH/tree/main |
| ReEvo | Code | MIT License | https://github.com/ai4co/reevo |
| HSEvo | Code | Available online | https://github.com/datphamvn/HSEvo |
| Synthetic problems | Dataset | Available Online | https://github.com/FeiLiu36/EoH/tree/main/examples/user_bo_caf |
| TSPLib | Dataset | Available for any non-commercial use | http://comopt.ifi.uni-heidelberg.de/software/TSPLIB95 |

