# OpenReview forum: "Monte Carlo Tree Search for Comprehensive Exploration in LLM-Based Automatic Heuristic Design"
_ICML.cc/2025/Conference — ICML 2025 poster_

### Official Review · Reviewer_AcSn · 2025-02-27

**Overall Recommendation:** 4

**Summary:**

This article discusses the limitations of population-based LLM-based automatic heuristic design, which makes it difficult to fully explore the heuristic space. This paper introduces MCTS for heuristic exploration and planning. Overall, in experiments, MCTS-AHD achieves significant performance advantages in various typical applications.

**Claims And Evidence:**

Generally clear and convincing.

**Essential References Not Discussed:**

This paper has included prior related findings.

**Experimental Designs Or Analyses:**

Some baseline results are missed on some testing problems. I think it is convincing to introduce HSEvo in TSP-GLS and ACO. Meanwhile, according to Figure 3, it seems that ReEvo and HSEvo can run for construction-based TSP. Can you demonstrate the results of the construction heuristic designed by these two algorithms on TSP?

**Methods And Evaluation Criteria:**

Make sense.

**Other Comments Or Suggestions:**

1.In addition to the heuristic performance, heuristics' execution efficiency can be another metric. I wonder about the efficiency of algorithms designed by MCTS-AHD.

2.There is no clear definition of the Gap and highlights in Table 2. Please provide more descriptions.

3.Appendix F.3 well presents the significant lead of MCTS-AHD compared to EoH, and I believe this significance analysis will be a good supplement to the existing work. Can other baselines such as ReEvo and HSEvo be introduced to calculate the p-value?

4.The ``MCT Root`` in Figure 3 should be the ``MCTS Root``.

**Other Strengths And Weaknesses:**

As strengths:

1.Using MCTS in LLM-based AHD is reasonable and novel.

2.This paper conducts comprehensive evaluations of MCTS-AHD on both CO problems and non-CO problems. MCTS-AHD seems to have a wide range of application scenarios and can get outstanding performance on NP-hard problems with clear descriptions.

3.This paper is well-written and clearly discusses the proposed method MCTS-AHD.
﻿

There are no major weaknesses in this paper, Please refer to **Experimental Designs Or Analyses**, **Other Comments Or Suggestions**, and **Questions For Authors** for my concerns and questions.

**Questions For Authors:**

1.The Q value design in MCTS is important. Can you conduct another ablation study on the normalization of the Q value in MCTS-AHD?

2.This paper only involves GPT-4o-mini and GPT-3.5-turbo for OpenAI GPT. Will more advanced GPT models perform better on MCTS-AHD?

3.The MCTS you are using seems to have structural differences from LLM reasoning. Can you further explain why it does not include multi-step exploration processes such as self-evaluation or rollback?

**Relation To Broader Scientific Literature:**

Ideas: Using MCTS to better explore and plan heuristics is an interesting and reasonable idea.

Results: Generally achieves better results on plenty of CO and non-CO problems compared to recent methods.

**Theoretical Claims:**

This paper does not include theoretical claims.

---

> ### Author Rebuttal · Authors · 2025-03-29
>
> Thank you so much for your time and effort in reviewing our work. We are glad to know that you find the proposed MCTS-AHD achieves significant performance advantages in various typical applications, using MCTS in LLM-based AHD is reasonable and novel and this paper is well-written.
>
> We address your concerns as follows.
>
> >**Experimental Designs Or Analyses. Baselines:**
>
> Thanks for your suggestions. We have complemented the ReEvo and HSEvo results in TSP-GLS and ACO TSP as follows.
> |Methods|TSP-ACO$N$=50|TSP-ACO$N$=100|CVRP-ACO$N$=50|CVRP-ACO$N$=100|MKP-ACO$N$=100|MKP-ACO$N$=200|BPP-ACO$N$=500|BPP-ACO$N$=1000|
> |:-|:-:|:-:|:-:|:-:|:-:|:-:|:-:|:-:|
> |DeepACO|0.71%|1.26%|**0.00%**|**0.00%**|0.79%|1.20%|**0.00%**|**0.00%**|
> |HSEvo|0.14%|0.50%|5.19%|9.04%|**0.00%**|0.01%|1.20%|1.38%|
> |MCTS-AHD(Ours)|**0.00%**|**0.00%**|4.48%|5.70%|0.03%|**0.00%**|0.48%|0.53%|
>
> |||TSP-GLS||
> |:-|:-:|:-:|:-:|
> |Methods|$N$=100|$N$=200|$N$=500|
> |KGLS|**0.003%**|0.227%|0.958%|
> |HSEvo|0.006%|0.223%|0.972%|
> |MCTS-AHD(Ours)|0.006%|**0.211%**|**0.949%**|
>
> |||TSP-Construction||
> |:-|:-:|:-:|:-:|
> |Methods|N=50|N=100|N=200|
> |POMO|**0.39%**|**3.01%**|20.45%|
> |HSEvo|12.70%|14.32%|15.68%|
> |MCTS-AHD(Ours)|9.69%|11.79%|**13.19%**|
>
> As the results above show, on almost all these test sets, MCTS-AHD can demonstrate advantages compared to HSEvo and ReEvo.
>
> >**Other Comments Or Suggestions 1. Heuristics' efficiency:**
>
> MCTS-AHD employs the MCTS to comprehensively explore the heuristic space. Therefore, compared to population-based baselines, MCTS-AHD will design more complex and inefficient heuristic algorithms with superior performance. As a future work, we will consider introducing the idea of multi-objective LLM-based AHD [1] to balance the efficiency and performance of heuristics.
>
> >**Other Comments Or Suggestions 2. Clear descriptions and typos:**
>
> Thanks for your suggestion. The Gap refers to the distance between the best-performing algorithm in the current dataset. In each dataset, the best-performing LLM-based AHD will be highlighted.
>
> >**Other Comments Or Suggestions 3. p-values for HSEvo and ReEvo:**
>
> Thanks for your suggestions. We have complemented the ReEvo and HSEvo results in Constructive TSP, TSP ACO, and MKP ACO for p-values as follows (the computation of avg, std, and p-value is the same as the Appendix F.3.). Results show that in designing constructive heuristics for TSP, MCTS-AHD can demonstrate clear superiority (with more than 90\% significance) compared to EoH, ReEvo, and HSEvo. In designing ACO heuristics for TSP and MKP, MCTS-AHD can demonstrate clear superiority to ReEvo and slight advantages to HSEvo.
>
>
> |COProblem|Methods|avg|std|p-value|
> |-|-|-|-|-|
> |**General**|**Framework:**|**Step-by-step**|**Construction**||
> |TSP50|EoH|6.386|0.080|**0.002855655**|
> ||ReEvo|6.400|0.128|**0.009529994**|
> ||HSEvo|6.397|0.069|**0.000796727**|
> ||MCTS-AHD|**6.280**|0.071||
> |**General**|**Framework:**|**ACO**|||
> |TSP50|EoH|5.828|0.003|**0.039230447**|
> ||ReEvo|5.844|0.029|**0.02477005**|
> ||HSEvo|5.819|0.018|0.115934978|
> ||MCTS-AHD|**5.795**|0.036||
> |MKP100|EoH|23.199|0.083|**0.037268388**|
> ||ReEvo|23.258|0.048|0.20630356|
> ||HSEvo|23.277|0.008|0.433650083|
> ||MCTS-AHD|**23.279**|0.025||
>
> >**Question 1. Ablation on normalization:**
>
> To demonstrate the importance of normalization, we present an additional ablation experiment. According to the table below where values are the optimality gaps, the normalization in MCTS-AHD demonstrates high importance.
>
> ||TSP50|KP100|
> |-|-|-|
> |MCTS-AHD(10runs)|10.661%|0.059%|
> |*w/o* Normalization|11.977%|0.083%|
>
>
> >**Question 2. Advanced GPT:**
>
> We complement the results of MCTS-AHD in designing constructive heuristics for TSP and KP with advanced GPT LLMs as follows. Values in the table are gaps to the optimal. Advanced GPT LLMs cannot produce superior performance and GPT-4o-mini may be the best choice for GPT LLMs.
>
> |Method|TSP$N=50$|TSP$N=100$|KP$N=100$|KP$N=200$|
> |-|:-:|:-:|:-:|:-:|
> |MCTS-AHD(GPT-4o-mini)|**9.69%**|11.79%|**0.05%**|**0.04%**|
> |MCTS-AHD(GPT-4o)|10.24%|**11.69%**|0.08%|0.10%|
> |MCTS-AHD(GPT-4)|10.35%|12.22%|0.10%|0.09%|
>
> >**Question 3. MCTS structure:**
>
> Thanks for your comment. Generally, MCTS-based LLM reasoning methods involve self-evaluation or rollout to evaluate a better quality value $Q$ of each state, but these methods may lead to biased evaluation results [2]. Optimization tasks provide reliable performance scores for the $Q$ value, so we believe MCTS-AHD does not need self-evaluations or rollouts for $Q$ value estimation.
>
> >**Reference**
>
> [1] Yao, Shunyu, et al. "Multi-objective evolution of heuristic using large language model." arXiv preprint arXiv:2409.16867 (2024).
>
> [2] Xu, Wenda, et al. "Pride and prejudice: LLM amplifies self-bias in self-refinement." arXiv preprint arXiv:2402.11436 (2024).

---

> > ### Comment · Reviewer_AcSn · 2025-04-02
> >
> > Thanks, it is clear now. I will raise my score.

---

> > > ### Author Response · Authors · 2025-04-02
> > >
> > > Thanks for your support and raising the score.

---

### Official Review · Reviewer_9L3t · 2025-03-07

**Overall Recommendation:** 4

**Summary:**

This paper introduces MCTS-AHD, a novel method that leverages MCTS to enhance the evolution of heuristic functions generated by LLMs for solving complex optimization tasks. The key contributions include the use of MCTS to organize and evolve heuristics in a tree structure, allowing for comprehensive exploration of the heuristic space and avoiding local optima. The method employs a set of LLM-based actions, including initialization, mutation, crossover, and tree-path reasoning, to iteratively refine heuristic functions. Experiments across various NP-hard combinatorial optimization problems and Bayesian Optimization tasks demonstrate that MCTS-AHD outperforms existing LLM-based AHD methods and handcrafted heuristics.

## update after rebuttal

I have raised my score to accept.

**Claims And Evidence:**

The claims made in the submission are supported by clear and convincing evidence. The authors provide extensive experimental results across multiple tasks, showing that MCTS-AHD consistently outperforms existing methods. The use of MCTS to explore the heuristic space and the proposed LLM-based actions are well-justified and demonstrated through both qualitative and quantitative analyses. The results indicate that MCTS-AHD can escape local optima and develop high-quality heuristics.

**Essential References Not Discussed:**

The paper provides a thorough review of related work. However, it might benefit from discussing recent advancements in the application of reinforcement learning techniques for heuristic optimization, which could provide additional context for the proposed method.

**Experimental Designs Or Analyses:**

The experimental designs are sound and valid. The authors conduct experiments on multiple datasets and tasks, comparing MCTS-AHD against various baselines, including handcrafted heuristics, neural combinatorial optimization methods, and existing LLM-based AHD methods. The results are statistically significant and demonstrate the effectiveness of MCTS-AHD. The authors also provide ablation studies to validate the necessity of the proposed components and actions.

**Methods And Evaluation Criteria:**

The proposed methods and evaluation criteria are appropriate for the problem at hand. The authors use a variety of NP-hard combinatorial optimization problems and Bayesian Optimization tasks to evaluate MCTS-AHD, ensuring that the method is tested across diverse scenarios. The evaluation criteria, including optimality gaps and performance improvements, are standard and relevant for these tasks.

**Other Comments Or Suggestions:**

The paper is well-written and presents its contributions clearly. A minor suggestion would be to include a discussion on the potential applications of MCTS-AHD in other domains to further highlight its significance.

**Other Strengths And Weaknesses:**

* Strengths:

1. The proposed method demonstrates significant improvements over existing LLM-based AHD methods and handcrafted heuristics.

2. The use of MCTS for heuristic evolution is novel and provides a promising direction for future research.

3. The paper includes comprehensive experiments and ablation studies, which validate the effectiveness of the proposed method and its components.

* Weaknesses:

1. The convergence speed of MCTS-AHD could be improved. Future work could explore hybrid methods combining MCTS with population-based approaches to enhance efficiency.

2. The paper does not provide detailed discussions on the computational complexity and scalability of MCTS-AHD for very large-scale optimization problems.

**Questions For Authors:**

1. How does the performance of MCTS-AHD compare to recent reinforcement learning methods like DeepSeek-R1-Zero, which have shown promise in enhancing LLMs for complex tasks?

2. Can the authors provide a detailed comparison of the time and computational resource requirements of MCTS-AHD versus previous population-based methods? This would help in understanding the trade-offs involved in using MCTS for heuristic evolution.

**Relation To Broader Scientific Literature:**

The key contributions of the paper are well-related to the broader scientific literature. The use of MCTS for heuristic evolution is novel and builds upon existing work in evolutionary algorithms, combinatorial optimization, and LLM-based heuristic design. The paper cites relevant prior work and clearly positions its contributions in the context of existing research.

**Theoretical Claims:**

The paper does not present any formal theoretical claims or proofs. The focus is on the empirical evaluation of the proposed method.

---

> ### Author Rebuttal · Authors · 2025-03-31
>
> Thank you so much for your time and effort in reviewing our work. We are glad to know that you find the proposed method demonstrates significant improvements and the paper includes comprehensive experiments and ablation studies.
>
> We address your concerns as follows.
>
> >**Weakness 1. Convergence speed:** The convergence speed of MCTS-AHD could be improved. Future work could explore hybrid methods combining MCTS with population-based approaches to enhance efficiency.
>
> Thanks for your suggestion. Yes, as listed in the Conclusion, future work could explore hybrid methods combining MCTS with population-based approaches to enhance the convergence speed of MCTS-AHD.
>
>
> >**Weakness 2. Computational complexity and scalability:** The paper does not provide detailed discussions on the computational complexity and scalability of MCTS-AHD for very large-scale optimization problems.
>
> Thank you very much for your comment. We would like to clarify that, unlike learning-based methods, heuristics designed on small-scale optimization instances often have high generalization abilities for large-scale optimization tasks [1]. As shown in the table below where constructive TSP heuristics designed by MCTS-AHD can well generalize to very large-scale instances with lower objective values.
>
> |Scale|N=500|N=1,000|N=2,000|N=5,000|
> |-|:-:|:-:|:-:|:-:|
> |Nearest Neighboor|20.2259|28.5305|39.2111|64.3547|
> |Construction heuristic designed by MCTS-AHD|18.7596|26.4281|37.2608|59.1260|
>
> Besides effectiveness, only heuristics with low time complexity can be easily applied to very large-scale instances. Current MCTS-AHD and other LLM-based AHD baselines do not include mechanisms to confine the algorithm complexity. To achieve trade-offs between heuristic algorithm efficiency and effectiveness, we should consider multi-objective LLM-based AHD methods [2]. We will consider this as future work and provide this discussion in our manuscript.
>
> >**Suggestion. Potential applications of MCTS-AHD in other domains:**
>
> Thanks for your valuable suggestion. As an advanced LLM-based AHD method, as discussed in [3,4], we believe MCTS-AHD has the potential to be applied to design heuristics for optimization tasks, machine learning tasks (In response to ``Weakness 3 by Reviewer ywZb``, we show the application of MCTS-AHD on a policy optimization task Gym-CarMountain-v0), and some scientific discovery tasks. We will include this discussion in our manuscript.
>
> >**Question 1. Compare to recent reinforcement learning methods:**
>
> Thanks for your comment. Recent test-time scaling methods like reinforcement learning-based reasoning model Deepseek-r1 and Openai-o1 cannot directly solve the AHD task with their thinking process. As shown in the table below, we prompt Deepseek-r1 and o1-preview to design heuristics and report the best-designed heuristic over $100$ LLM calls. The advanced test-time scaling methods can only find low-quality heuristics, which demonstrates the significance of using MCTS-AHD in AHD tasks.
>
>
> |Task|TSP-$N$=50||TSP-$N$=100||KP-$N$=100,$W$=25||KP-$N$=200,$W$=25||
> |-|:-:|:-:|:-:|:-:|:-:|:-:|:-:|:-:|
> |Method|Obj.↓|Gap|Obj.↓|Gap|Obj.↑|Gap|Obj.↑|Gap|
> |Optimal|5.675|-|7.768|-|40.271|-|57.448|-|
> |Deepseek-r1 (100 trails)|6.916|21.85%|9.595|23.52%|40.225|0.12%|57.395|0.09%|
> |o1-preview (100 trails)|6.959|22.62%|9.706|24.94%|40.225|0.12%|57.395|0.09%|
> |MCTS-AHD(GPT-4o-mini)|**6.225**|**9.69%**|**8.684**|**11.79%**|**40.252**|**0.05%**|**57.423**|**0.04%**|
> |MCTS-AHD(DeepSeek-v3)|6.348|11.85%|8.859|14.04%|40.233|0.10%|57.402|0.08%|
>
>
> >**Question 2. Time and computational resource requirements:**
>
> Thanks for your suggestion. As shown in the table below, we provide a detailed comparison of the time and token consumption of MCTS-AHD and population-based baselines in five application scenarios. We calculate the token number based on GPT-4o-mini. Results demonstrate that compared to population-based methods, MCTS-AHD has a slight efficiency decrease and maintains a similar level of token consumption compared to LLM-based AHD baselines.
>
> |Methods|Consumption|Construction-TSP|Construction-KP|ACO-TSO|ACO-MKP|BO-CAF|
> |-|-|:-:|:-:|:-:|:-:|:-:|
> |**EoH**|Time|2h|2h|4h|4h|15h|
> ||InputToken|0.8M|0.7M|1M|1M|1.2M|
> ||OutputToken|0.2M|0.2M|0.5M|0.5M|0.5M|
> |**ReEvo**|Time|2h|-|5h|5h|-|
> ||InputToken|1.1M|-|1.3M|1.3M|-|
> ||OutputToken|0.4M|-|0.6M|0.5M|-|
> |**MCTS-AHD**|Time|4h|3h|8h|4h|14h|
> ||InputToken|1M|1M|1.2M|1.3M|1.3M|
> ||OutputToken|0.3M|0.2M|0.5M|0.6M|0.6M|
>
> >**Reference:**
>
> [1] Liu, Fei, et al. "Algorithm evolution using large language model." arXiv preprint arXiv:2311.15249 (2023).
>
> [2] Yao, Shunyu, et al. "Multi-objective evolution of heuristic using large language model." arXiv preprint arXiv:2409.16867 (2024).
>
> [3] Liu, Fei, et al. "A systematic survey on large language models for algorithm design." arXiv preprint arXiv:2410.14716 (2024).
>
> [4] Liu, Fei, et al. "Llm4ad: A platform for algorithm design with large language model." arXiv preprint arXiv:2412.17287 (2024).

---

> > ### Comment · Reviewer_9L3t · 2025-04-02
> >
> > Thank you for your detailed response. While I appreciate the analysis on the quality of heuristics found by advanced test-time scaling methods such as Deepseek-r1 and Openai-o1, I believe it is also crucial to evaluate their performance in terms of time efficiency and token consumption. Specifically, I would like to see a comparison of these metrics between MCTS-AHD and the advanced test-time scaling methods (Deepseek-r1 and Openai-o1) . This comparison would provide a more comprehensive understanding of the trade-offs involved in using MCTS-AHD versus these other methods.

---

> > > ### Author Response · Authors · 2025-04-02
> > >
> > > Thank you very much for your valuable comment. We collect time and token consumptions of test time scaling methods (i.e., Deepseek-r1 and Openai-o1) in AHD tasks as follows. Since the specific tokenizer for Openai-o1 is closed-source, we use the tokenizer of GPT-4o-mini to obtain all the token consumptions.
> > >
> > > | Method | Consumption | Construction-TSP| Construction-KP|
> > > |:-:|-|:-:|:-:|
> > > | MCTS-AHD (1,000 GPT-4o-mini code generations) | Time | 4h|3h|
> > > | | Input Token| 1M|1M|
> > > | | Output Token |0.3M | 0.2M |
> > > | Deepseek-r1 (per trial) | Time |0.92m/trial|2.65m/trial |
> > > || Input Token|0.15K/trial|0.16K/trial |
> > > | | Output Token | 2.3K/trial|1.0K/trial|
> > > | Openai-o1 (per trial)| Time |0.279m/trial | 0.375m/trial |
> > > | | Input Token|0.15K/trial|0.16K/trial |
> > > | | Output Token |0.14K/trial|0.14K/trial |
> > >
> > >
> > > In designing step-by-step construction heuristics for KP as an example, referring to the price of GPT-4o mini, the algorithm design for MCTS-AHD requires approximately \\$ 0.3. Compared to MCTS-AHD, according to the official prices of Deepseek-r1 and Openai-o1, obtaining approximately 150 trials with Deepseek-r1 and 30 trials with Openai-o1 will cost the same budget (\\$ 0.3). Considering the time consumption, Openai-o1 has a relatively fast response, which can generate 500 trials within the time consumption of MCTS-AHD.
> > >
> > > Therefore, we believe that Deepseek-r1 or Openai-o1 are more feasible for obtaining relatively low-quality heuristic algorithms with limited cost and time budgets while LLM-based AHD algorithms such as MCTS-AHD can be better choices for better-performing heuristic algorithms.

---

### Official Review · Reviewer_ywZb · 2025-03-14

**Overall Recommendation:** 3

**Summary:**

This paper introduces MCTS-AHD, a Monte Carlo Tree Search (MCTS)-based method for automatic heuristic design (AHD) using Large Language Models (LLMs). MCTS-AHD organizes heuristics in a tree structure to enable more comprehensive exploration and refinement. Ultimately, the goal is to generate more efficient, robust, and generalizable heuristic functions for combinatorial optimization (CO) and Bayesian optimization (BO) tasks, enhancing the ability to solve complex optimization problems.

**Claims And Evidence:**

Yes

**Essential References Not Discussed:**

The paper has included the main related works that are crucial for understanding the context.

**Experimental Designs Or Analyses:**

I have checked all the experiments in the experimental section.

**Methods And Evaluation Criteria:**

Yes

**Other Comments Or Suggestions:**

See weaknesses

**Other Strengths And Weaknesses:**

Strengths:

1. The topic that leveraging LLMs to automatically enhance heuristics is intresting.
2. The paper is well-written.

Weaknesses:

1. There is no formal convergence proof, making it unclear whether MCTS-AHD is guaranteed to consistently find optimal or near-optimal heuristics across different search spaces. Also, there is no computational complexity analysis, which is essential to understand the efficiency and scalability of the approach compared to traditional methods.
2. The method relies heavily on LLMs for both generating and refining heuristics, making its performance sensitive to the quality and stability of LLM-generated outputs.
3. It is recommended to extend the evaluation to other types of optimization problems, such as code search, to assess the generalizability of the method across different domains.
4. MCTS-AHD requires numerous LLM evaluations during its tree search, leading to higher computational costs compared to traditional population-based methods, which might limit practical applicability in resource-constrained environments.

**Questions For Authors:**

See weaknesses

**Relation To Broader Scientific Literature:**

The key contributions of MCTS-AHD are situated at the combination of automatic heuristic design (AHD), Monte Carlo Tree Search (MCTS), and Large Language Models (LLMs). The paper extends prior research in these domains while addressing specific shortcomings of existing approaches.

**Theoretical Claims:**

The paper does not provide theoretical claim.

---

> ### Author Rebuttal · Authors · 2025-03-31
>
> Thank you so much for your time and effort in reviewing our work. We are glad to know that you find MCTS-AHD addressing specific shortcomings of existing approaches and the paper is well-written.
>
> We address your concerns as follows.
>
> >**Weakness 1. Computational complexity analysis:** There is no computational complexity analysis, which is essential to understand the efficiency and scalability of the approach compared to traditional methods.
>
> Thank you for your valuable comment. The time consumption of running MCTS-AHD is from three parts: evaluating the performance of heuristic algorithms, employing LLMs for heuristic function generation, and employing MCTS to select heuristics. The time complexity of the first two parts cannot be represented explicitly. The time consumption of heuristic evaluation is related to the complexity of generated heuristics and the size of the evaluation dataset $D$, while the time consumption of LLM generation is related to the selection of LLMs.
>
> For the third part, the total time complexity of the heuristic selection in MCTS-AHD with $T$ evaluations is $\mathcal{O}(HT)$, where H is the maximal height of MCTS. The space complexity is $\mathcal{O}(T)$ for preserving all LLM-generated heuristics. When maintaining a $M$-size population, the time complexity of heuristic selection in EoH and ReEvo is $\mathcal{O}(MT)$, and their space complexity is $\mathcal{O}(M)$.
>
> >**Weakness 1 & 2. Stability and Convergence proof:** The method relies heavily on LLMs for both generating and refining heuristics, making its performance sensitive to the quality and stability of LLM-generated outputs. There is no formal convergence proof.
>
> Thanks for your insightful comment. We agree that the effectiveness of all the LLM-based AHD methods largely depends on the stability and ability of LLMs to improve heuristics. There is no formal convergence proof for these methods.
>
>
> >**Weakness 3. Other types of optimization problems, such as code search:** It is recommended to extend the evaluation to other types of optimization problems, such as code search.
>
> Thanks for your comments. As an AHD method, MCTS-AHD aims at designing and evolving high-quality heuristics for optimization problems within a given solving framework. Code search [1,2] is indeed a typical optimization problem. However, there are no compelling heuristics to solve the code search tasks so MCTS-AHD cannot be directly applied for code searches. When directly implementing MCTS-AHD to code search (e.g., the APPS dataset [1]), the current MCTS-AHD cannot handle code bugs and becomes ineffective.
>
> |Methods |EoH|ReEvo|MCTS-AHD|
> |-|:-:|:-:|:-:|
> |Gym-MountainCar-v0|140.3|117.6|**115.0**|
>
> To further assess the generalizability of MCTS-AHD across different domains. We evaluate MCTS-AHD on a policy optimization task ``MountainCar-v0`` based on the Gym framework. The table above shows the average steps needed to reach the goal where the heuristic policy designed by MCTS-AHD is more effective. We run AHD methods three times for average performances and take GPT-4o-mini as LLMs.
>
> In conclusion, MCTS-AHD demonstrates superiority across combinatorial optimization, Bayesian optimization, and policy optimization using the same set of hyperparameters, so we believe MCTS-AHD can generalize to a wide range of application scenarios.
>
> >**Weakness 4. Numerous LLM evaluations:** MCTS-AHD requires numerous LLM evaluations during its tree search.
>
> Thanks for your comment. We agree that due to the proposed thought-alignment procedure, MCTS-AHD requires more LLM calls. In each heuristic design with $T$ performance evaluations, MCTS-AHD conducts $2T$ LLM calls (includes $T$ LLM calls for code generation and $T$ LLM calls for thought-alignment procedures), while EoH requires $T$ LLM calls and ReEvo requires $\frac{22}{15}T$ LLM calls in this case.
>
> However, we want to clarify that the LLM calls for thought-alignment procedures only consume a limited amount of tokens. As shown in the table below, we calculate the total token costs of MCTS-AHD and baselines on a series of representative heuristics design scenarios (setting $T=1000$ and using GPT-4o-mini as LLMs). The results show that MCTS-AHD will not make significantly higher token costs while achieving better heuristic performances.
>
> |Methods|Consumption|Construction-TSP|Construction-KP|ACO-TSO|ACO-MKP|BO-CAF|
> |-|-|:-:|:-:|:-:|:-:|:-:|
> |**EoH**|InputToken|0.8M|0.7M|1M|1M|1.2M|
> ||OutputToken|0.2M|0.2M|0.5M|0.5M|0.5M|
> |**ReEvo**|InputToken|1.1M|-|1.3M|1.3M|-|
> ||OutputToken|0.4M|-|0.6M|0.5M|-|
> |**MCTS-AHD**|InputToken|1M|1M|1.2M|1.3M|1.3M|
> ||OutputToken|0.3M|0.2M|0.5M|0.6M|0.6M|
>
> >**Reference:**
>
> [1] Zhong, Li, Zilong Wang, and Jingbo Shang. "Debug like a human: A large language model debugger via verifying runtime execution step-by-step." arXiv preprint arXiv:2402.16906 (2024).
>
> [2] Hendrycks D, Basart S, Kadavath S, et al. Measuring coding challenge competence with apps[J]. arXiv preprint arXiv:2105.09938, 2021.

---

### Official Review · Reviewer_x4g3 · 2025-03-15

**Overall Recommendation:** 4

**Summary:**

This paper introduces MCTS-AHD, which integrates MCTS into LLM-based AHD to improve heuristic search exploration. MCTS-AHD organizes heuristics in a tree structure, allowing for deeper refinement of temporarily weaker candidates. Key techniques include progressive widening, exploration decay, and tree-path reasoning, enabling more comprehensive heuristic evolution. Experiments on NP-hard optimization tasks (e.g., TSP, KP) and Bayesian optimization demonstrate superior performance over handcrafted heuristics and existing AHD methods.

**Claims And Evidence:**

The paper’s primary claims are backed up by reasonably thorough empirical results on multiple NP-hard problems (TSP, KP, BPP, CVRP, etc.). The authors also present ablation studies demonstrating the contributions of individual components (e.g., progressive widening, exploration decay) in MCTS-AHD.

**Essential References Not Discussed:**

Not found

**Experimental Designs Or Analyses:**

Yes I check the validity of experimental designs and analyses.

**Methods And Evaluation Criteria:**

Yes.

**Other Comments Or Suggestions:**

page 6 line 298  LMM - LLM

**Other Strengths And Weaknesses:**

The paper is very well written and introduce a effective method for overcoming local optima. The experiments are detailed. The approach relies on LLM-generated code, which may sometimes produce heuristics that are either non-executable or logically flawed. Although the paper introduces a thought-alignment process to mitigate this issue, further analysis on the robustness of LLM outputs and the effectiveness of error handling would be beneficial.

**Questions For Authors:**

The paper mentions running time and token cost for the KP task, similar information is missing for other tasks. Could the authors provide more details for those tasks?

**Relation To Broader Scientific Literature:**

This paper is primarily an extension of previous work on LLM for AHD, further incorporating the MCTS algorithm to explore the space of heuristic functions, thereby addressing known limitations of earlier methods.

**Theoretical Claims:**

Yes. I checked the standard MCTS-related proofs.

---

> ### Author Rebuttal · Authors · 2025-03-30
>
> Thank you so much for your time and effort in reviewing our work. We are glad to know that you find the proposed MCTS-AHD is an effective method for overcoming local optima, the paper is very well written, and the experiments are detailed.
>
> We address your concerns as follows.
>
> > **Weakness. Robustness of LLM output and error handling:** The approach relies on LLM-generated code, which may sometimes produce heuristics that are either non-executable or logically flawed. Although the paper introduces a thought-alignment process to mitigate this issue, further analysis on the robustness of LLM outputs and the effectiveness of error handling would be beneficial.
>
>
> Thank you for your insightful comment. We agree that LLMs may sometimes produce non-executable code. However, it's worth noting that unlike tasks involving code generation from scratch [1,2], where debugging plays a crucial role, LLM-based AHD methods only design the key heuristic function within predefined solving frameworks. This setup significantly reduces the likelihood of generating invalid codes [3]. Empirically, when using GPT-4o-mini, the probability of MCTS-AHD generating invalid code for step-by-step construction TSP heuristics is only about 15\%–20\%. Therefore, we believe that LLM-based debugging for AHD is not indispensable. To mitigate the potential negative impact of invalid codes on AHD, MCTS-AHD discards these invalid heuristic function codes and relies on conducting numerous LLM calls for valid code samples. We will add this discussion to our manuscript.
>
>
>
> > **Question. Time and token costs for other tasks:** The paper mentions running time and token cost for the KP task, similar information is missing for other tasks. Could the authors provide more details for those tasks?
>
> Thanks for your comment. As shown in the table below, we provide more statistics on the time and token consumption of MCTS-AHD and the baselines in more application scenarios. We calculate the token numbers based on using GPT-4o-mini as LLMs. Compared to LLM-based AHD baselines, MCTS-AHD demonstrates a slight efficiency decrease and maintains a similar level of token consumption.
>
> |Methods|Consumption|Construction-TSP|Construction-KP|ACO-TSO|ACO-MKP|BO-CAF|
> |-|-|:-:|:-:|:-:|:-:|:-:|
> |**EoH**|Time|2h|2h|4h|4h|15h|
> ||InputToken|0.8M|0.7M|1M|1M|1.2M|
> ||OutputToken|0.2M|0.2M|0.5M|0.5M|0.5M|
> |**ReEvo**|Time|2h|-|5h|5h|-|
> ||InputToken|1.1M|-|1.3M|1.3M|-|
> ||OutputToken|0.4M|-|0.6M|0.5M|-|
> |**MCTS-AHD**|Time|4h|3h|8h|4h|14h|
> ||InputToken|1M|1M|1.2M|1.3M|1.3M|
> ||OutputToken|0.3M|0.2M|0.5M|0.6M|0.6M|
>
> > **Other Comments Or Suggestions. Typo:** page 6 line 298 LMM - LLM problems.
>
> Thank you for pointing out the typos. We will correct typos accordingly.
>
>
> > **References**
>
> [1] Zhong, Li, Zilong Wang, and Jingbo Shang. "Debug like a human: A large language model debugger via verifying runtime execution step-by-step." arXiv preprint arXiv:2402.16906 (2024).
>
> [2] Hendrycks D, Basart S, Kadavath S, et al. Measuring coding challenge competence with apps[J]. arXiv preprint arXiv:2105.09938, 2021.
>
> [3] Liu, Fei, et al. "A systematic survey on large language models for algorithm design." arXiv preprint arXiv:2410.14716 (2024).

---

### Decision · Program_Chairs · 2025-05-01

**Decision:**

Accept (poster)

**Comment:**

This paper proposed a Monte Carlo Tree Search method to enhance LLM based Automatic Heuristic Design. Reviewers acknowledged the novelty and good empirical performance of the proposed method. Several concerns are raised, including missing discussions on the code generation error and efficiency of the generated heuristics. All reviewers are positive, but three of them seems not very familiar with this specific topic.